# `GMAgent`: A Graph-oriented Multi-agent Collaboration Framework for Text-attributed Graph Analysis

**Hang Lv** *lvhangkenn@gmail.com*
*Fuzhou University, Fuzhou, China*

**Pengxiang Zhan** *yyyzhanpengxiang@163.com*
*Fuzhou University, Fuzhou, China*

**Yanchao Tan** *yctan@fzu.edu.cn*
*Fuzhou University, Fuzhou, China*

**Zixuan Guo** *832304221@fzu.edu.cn*
*Fuzhou University, Fuzhou, China*

**Shiping Wang** *shipingwangphd@163.com*
*Fuzhou University, Fuzhou, China*

**Carl Yang** *j.carlyang@emory.edu*
*Emory University, Atlanta, USA*

**Reviewed on OpenReview:** *https://openreview.net/forum?id=iBm79ePO ex*

## Abstract

Text-Attributed Graphs (TAGs) are crucial for modeling interconnected data in numerous real-world applications. Graph Neural Networks (GNNs) excel at efficiently capturing global structural information across TAGs, while Large Language Models (LLMs) offer strong capabilities in local semantic understanding. Despite the recent development of integrating GNNs and LLMs for TAG analysis, these approaches often fail to fully exploit their complementary strengths by relying primarily on a single architecture. Furthermore, LLM-based multi-agent collaboration systems have shown promising potential across diverse fields. However, their integration with GNNs for graph analytical tasks remains underexplored. To this end, we introduce `GMAgent`, a novel graph-oriented multi-agent collaboration framework that effectively and flexibly interacts between diverse GNN-based and LLM-based graph agents, facilitating comprehensive TAG analysis. First, we deploy multiple GNNs as graph agents to perform conflict evaluation, identifying conflict scenarios for further multi-agent collaboration. Then, we repurpose LLMs as graph agents via graph-driven instruction tuning and adopt a role-play expert recruiting strategy, thereby generating LLM graph experts' initial analyses for conflict scenarios. Finally, we conduct a graph-oriented multi-agent collaboration to effectively and efficiently guide collaborative self-reflection on graph experts and the final answer selection. Extensive experimental results on five datasets demonstrate significant improvements, showcasing the potential of our `GMAgent` in improving the effectiveness, interoperability, and flexibility of comprehensive TAG analysis.

## 1 Introduction

Text-Attributed Graphs (TAGs), where each node can be associated with textual attributes, are common across various real-world applications. Due to their rich semantics and complex structures, TAGs have been widely used in diverse domains, such as social networks, academic networks, e-commerce networks, and web

page analytics (Pan et al., 2024; Ren et al., 2024), supporting core tasks such as node classification and link prediction (Jin et al., 2024; Tan et al., 2025). To accurately handle TAGs, Graph Neural Networks (GNNs) have been commonly adopted for capturing the global structural information of graphs (Wang et al., 2024c; Zhu et al., 2021). However, GNNs often struggle to fully integrate the rich semantics embedded in textual attributes. Inspired by the successes of Large Language Models (LLMs), some researchers have explored LLMs for accurately capturing contextual semantics of graph attributes (Tang et al., 2024a; Fang et al., 2024; Huang et al., 2024a), while their limited input tokens hinder the processing of large-scale graphs. Although these efforts have been made to combine GNNs and LLMs, they generally rely on either GNNs or LLMs as the primary backbone, limiting their abilities to comprehensively exploit the strengths of both.

Recently, LLMs' advanced task understanding and self-planning capabilities have led to the development of LLM-based multi-agent collaboration systems across diverse fields (Chen et al., 2025; Fan et al., 2025; Tang et al., 2024b). Nevertheless, the integration of LLMs and GNNs within multi-agent collaboration frameworks for graph analytical tasks remains largely underexplored. Figure 1 illustrates the potential of a multi-agent collaboration framework that combines the global structural patterns captured by GNNs with the local semantic understandings provided by LLMs. In this multi-agent framework, GNNs and LLMs can work together as complementary backbones, learning from each other and collaborating to improve graph analytical tasks (e.g., node classification, link prediction, and graph classification). To achieve these goals, we further address the following three key technical challenges.

**Challenge I:** *How to effectively utilize GNNs as graph agents for interacting with other graph agents?* GNNs excel in capturing global structural information across the whole TAG via message-passing and aggregation mechanisms, especially providing a clear computational advantage in large-scale graph analysis. While deploying GNNs as graph agents can significantly enhance the understanding of TAGs, the inherent architectures of GNNs limit their interpretability when interacting with other graph agents.

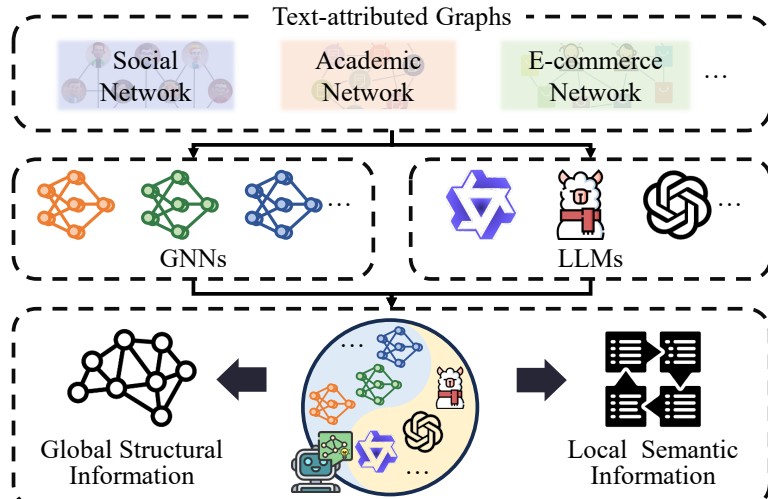

Figure 1: An illustrative toy example of the potential for a multi-agent collaboration framework that can simultaneously harness the global structural learning power of GNNs and the local semantic richness of LLMs for TAG analysis.

**Challenge II:** *How to repurpose LLMs as graph agents to accurately understand complex graph data and execute graph analytical tasks?* To fully exploit the potential of LLMs for graph analysis, an essential obstacle is the accurate understanding and reasoning capabilities of LLMs given graph data with abundant attributes and flexible structures. Despite LLMs' strengths in understanding semantics, they still struggle to precisely understand graph structures and the information needs of different graph analytical tasks.

**Challenge III:** *How to properly perform multi-agent collaboration with GNN-based and LLM-based graph agents for comprehensive TAG analysis?* The effectiveness of multi-agent collaboration largely depends on how to integrate the strengths of multiple agents to facilitate comprehensive TAG analysis. Balancing the global structural learning power of GNN-based graph agents with the local semantic comprehension of LLM-based graph agents presents a significant challenge in robustly improving the accuracy and efficiency of graph-oriented multi-agent collaboration.

To tackle these challenges, we propose a **G**raph-oriented **M**ulti-**Agent** collaboration framework for text-attributed graph analysis (`GMAgent`), which consists of three key steps: (i) *Deploying GNNs as Graph Agents*,

where we perform conflict evaluation based on multiple trained GNN models to identify conflicting scenarios for further multi-agent collaboration; (ii) *Repurposing LLMs as Graph Agents*, where we obtain an LLM-powered graph agent via graph-driven instruction tuning and adopt a role-play expert recruiting strategy, collecting LLM graph experts' initial analyses for conflicting scenarios; (iii) *Graph-oriented Multi-agent Collaboration*, where we enable advanced LLMs (e.g., GPT-4o) to assign a confidence score for each LLM graph expert and generate a summary report, along with all GNN and LLM experts' analyses, to guide collaborative self-reflection of LLM experts and final answer selection.

Our overall contributions are summarized as follows:

- *Formulation of the Graph-oriented Multi-agent Framework.* We establish a first multi-agent framework that effectively and flexibly engages interactions between diverse GNN-based and LLM-based graph agents, facilitating comprehensive TAG analysis.

- *Effective Model Designs.* We design and implement a set of models and mechanisms, including conflict evaluation, graph-driven instruction tuning, role-play expert recruiting, summary report generation, and collaborative self-reflection, constituting a robust graph-oriented multi-agent collaboration framework for comprehensive TAG analysis.

- *Extensive Experiments across Graph Analytical Tasks and Datasets.* We conduct thorough experiments to validate our proposed approach with five real-world datasets, demonstrating its superiority over existing state-of-the-art methods and highlighting its effective, interpretable, and flexible abilities in enhancing TAG analysis.

## 2 Related Work

### 2.1 Text-attributed Graph Analysis

Graph analysis methods, especially Graph Neural Networks (GNNs), have shown effectiveness in capturing structural information for graph data (Zeng et al., 2025; Wang et al., 2024c; Zhu et al., 2021). Traditional GNNs, such as Graph Convolutional Networks (GCNs) (Kipf & Welling, 2017) and Graph Attention Networks (GATs) (Veličković et al., 2018), are widely used for learning node and edge representations in graphs, excelling in extracting structural relationships. However, when it comes to processing rich textual attributes associated with nodes or edges, these models face limitations (Tan et al., 2023; Wang et al., 2024c; Fang et al., 2024; He et al., 2024; Zhao et al., 2023).

Recently, the emergence of Large Language Models (LLMs), with their robust semantic understanding, offers a way to incorporate textual information into Text-Attributed Graph (TAG) learning (Fang et al., 2024; Tan et al., 2023). For instance, GraphGPT (Tang et al., 2024a) demonstrated how LLMs can process and understand complex graph structures by combining text-based knowledge with graph structural information. Pan et al. (2024) distilled knowledge from LLMs for learning on TAGs. While LLMs excel at interpreting and representing text, they are insufficient for capturing the global structural patterns that GNNs handle so well, highlighting the need for a more integrated approach (Li et al., 2024).

Building on the complementary strengths of GNNs and LLMs, recent research has explored two main categories of integrated approaches, based on whether the backbone is a GNN or an LLM. In the first category, GNNs serve as the backbone, with LLMs providing additional semantic context. Methods such as ConGraT (Brannon et al., 2024) and GRENADE (Li et al., 2023) align node embeddings generated by GNNs with representations from LLMs, effectively combining the structural information from TAGs with the textual understanding of LLMs. In the second category, LLMs are used as the backbone, and GNNs are incorporated to infuse structural graph information into LLMs. Approaches like DGTL (Qin et al., 2023) and GraphAdapter (Huang et al., 2024b) co-train transformer layers in LLMs alongside graph neural layers, enabling LLMs to better capture the structural dependencies in TAGs. Despite the progress made by both categories, a key limitation remains: these models typically rely on either GNNs or LLMs as the primary backbone, which restricts their ability to fully leverage the strengths of both. As a result, they fail to simultaneously harness the structural learning power of GNNs and the semantic richness of LLMs.

## 2.2 Multi-agent Systems

Multi-agent systems have been explored across a wide range of applications and domains (Chen et al., 2025; Liu et al., 2024b; Chen et al., 2023), demonstrating their capability to efficiently handle complex tasks through agent coordination. For example, AutoGen (Wu et al., 2024a) has focused on automating multi-agent collaboration in diverse tasks, such as math problem-solving, group chat, and coding. BadAgents (Yang et al., 2024b) formulated a framework for agent backdoor attacks, which can introduce malicious behavior in the intermediate reasoning process while keeping the final output correct. These approaches demonstrate the potential of multi-agent systems to scale and efficiently solve problems through agent coordination.

In recent years, LLM-based multi-agent systems have emerged as a powerful tool, primarily divided into two categories: collaboration and competition (Wang et al., 2024b). The first category involves collaborative multi-agent systems, where multiple LLM-based agents work together toward a common goal. These systems have proven effective in areas like cooperative decision-making (Piatti et al., 2024), majority voting (Chen et al., 2024), medical domains (Fan et al., 2025; Tang et al., 2024b), and code generation (Islam et al., 2024). For example, GOVSIM (Piatti et al., 2024) represented the LLM's strategic interaction and cooperative decision-making capability. The second category includes adversarial multi-agent systems, where agents have conflicting objectives. These methods have been applied in competitive environments, such as games (Feng et al., 2024), debate (Liu et al., 2024c), and evaluation (Chan et al., 2024). For example, GroupDebate (Liu et al., 2024c) involved dividing all agents into multiple debate groups, enhancing the performance and efficiency in the multi-agent debate system.

Despite the wide-ranging success of multi-agent LLM systems in these fields, their application in graph analytical tasks remains relatively unexplored. In the graph domain, existing multi-agent systems often employ GNNs to support agent coordination structures or to model agent states (Zhang et al., 2024; Duan et al., 2024). For example, VillagerAgent (Dong et al., 2024) introduced a directed acyclic graph framework to manage task assignments and track agent states, while CAG-ODE (Huang et al., 2024c) employed a GNN as the ordinary differential equations function to model continuous agent interactions. GraphAgent-Reasoner (Hu et al., 2024) further extended LLM-based multi-agent collaboration for graph reasoning, though its application remains limited to this specific task. Wu et al. (2024b) made progress by integrating GNNs and LLMs within a multi-agent framework to enhance graph learning for task planning. As closest to us, MARK (Fu et al., 2025) utilized GNNs to identify uncertain nodes via graph clustering, and then generated ranking-based guidance through LLM-based multi-agent collaboration. However, MARK's reliance on clustering restricts its applicability to broader graph tasks like multi-label node classification or link prediction. Although both LLMs and GNNs have shown individual success in multi-agent systems, their integration for graph analytical tasks is still limited, leaving significant potential for deeper investigation and development in this area.

# 3 The `GMAgent` Framework

## 3.1 Problem Formulation and `GMAgent` Overview

Given Text-Attributed Graphs (TAG) $\mathcal{G}$ and multiple graph analytical $\tau_i$ (e.g., node classification $\tau_1$), the goal of `GMAgent` is to develop a multi-agent collaboration framework that flexibly integrates the strengths of various Graph Neural Network (GNN) graph agents $\{GNN_g\}_{g=1}^{M_G}$ and Large Language Model (LLM) graph agents $\{LLM_l\}_{l=1}^{M_L}$, improving graph analytical task performances on $\tau_i$.

To achieve this goal, we first deploy multiple GNNs trained on the specific TAG dataset as graph agents, performing conflict evaluation to identify conflict scenarios for multi-agent collaboration. Then, we generate an LLM-powered graph agent via graph-driven instruction tuning, integrating CoT-based instructions from advanced LLMs (e.g., GPT-4o) with task-specific instructions. Adopting a role-play expert recruiting strategy, diverse LLM graph experts provide initial analyses for conflict scenarios. Finally, we introduce a graph-oriented multi-agent collaboration between GNN and LLM experts, where advanced LLMs (e.g., GPT-4o) assign a confidence score to each LLM expert and generate a summary report to guide collaborative self-reflection on LLM experts and final answer selection. The overall framework of our `GMAgent` is shown in Figure 2.

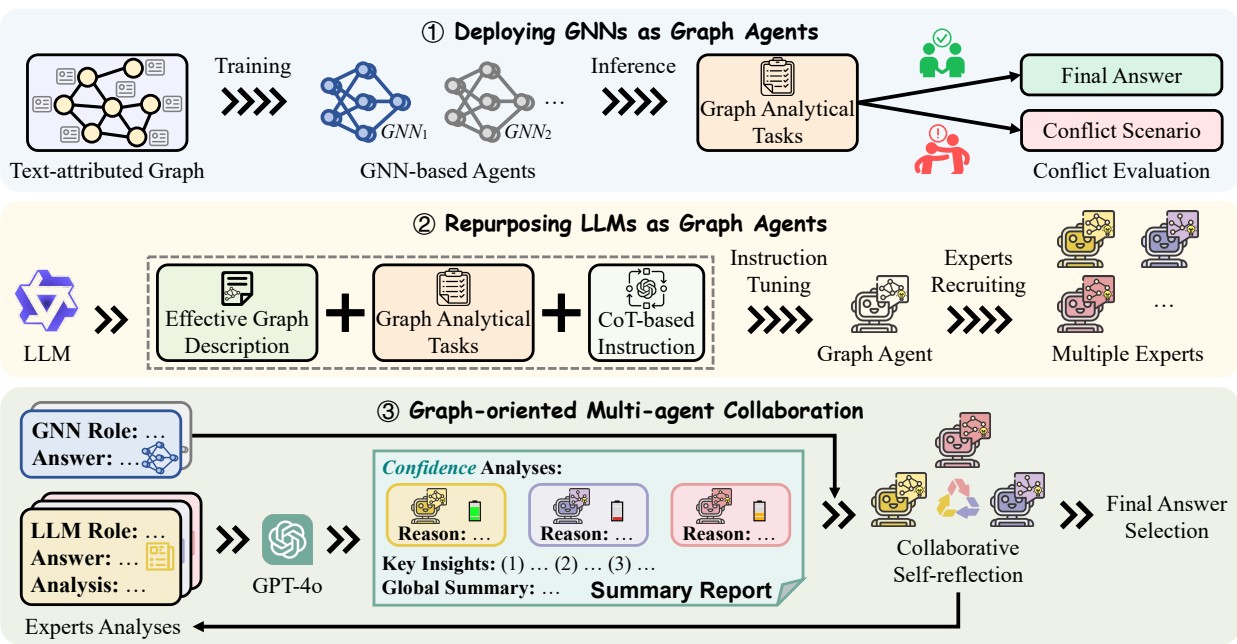

Figure 2: The overall of our `GMAgent` framework, consisting of three key steps, which are ① Deploying GNNs (e.g., GCN and GAT) as Graph Agents, ② Repurposing LLMs (e.g., Qwen2-7B) as Graph Agents, and ③ Graph-oriented Multi-agent Collaboration.

## 3.2 Deploying GNNs as Graph Agents

GNNs demonstrate strong performance on multiple training tasks and datasets (Luo et al., 2024; 2025). They effectively enhance node representations with structural information via message-passing and aggregation patterns, considering the global information across the whole TAG (Pan et al., 2024; Wang et al., 2024c; Zhu et al., 2021). As shown in Figure 3(a), GNN models, like GCN (Kipf & Welling, 2017) and GAT (Veličković et al., 2018), can achieve the average accuracy rates of 73.08% and 73.79%, respectively. Notably, when both models provide consistent predictions (i.e., "GCN + GAT"), the accuracy improves further, demonstrating that GNNs are capable of making reliable and accurate inferences. In addition to their accuracy, GNNs offer a clear computational advantage when dealing with large-scale graphs. Compared with LLMs that accurately capture contextual semantic understanding within limited input tokens, GNNs can efficiently process large-scale graph structures utilizing their architecture's inherent parallel message-passing mechanism (Yang et al., 2020a). As shown in Figure 3(b), we compare the inference efficiency (seconds per response) of GCN, LLaMA3-8B (Dubey et al., 2024), and Qwen2-7B (Yang et al., 2024a) across three datasets, where LLMs like LLaMA3-8B and Qwen2-7B exhibit significantly higher inference times than GNNs due to producing interpretable answers.

To this end, we propose to deploy GNNs as graph agents within our multi-agent framework for TAG analysis, which balances effectiveness and computational efficiency. Specifically, we first train multiple GNNs (e.g., GCN and GAT) tailored for specific datasets and graph analytical tasks (e.g., node classification). Then, we introduce a conflict evaluation to identify conflict scenarios based on each GNN-based graph agent's inference results. These results for conflict scenarios are used for further multi-agent collaboration.

### 3.2.1 GNN-based Graph Agents Training

To obtain the GNN-based graph agent, we train each GNN on the TAG dataset $\mathcal{G} = (\mathcal{V}, \mathcal{E}, \mathcal{X}, \mathcal{S})$ for a graph analytical task $\tau_i$. Each node $v_n \in \mathcal{V}$ has an initial node embedding, $\boldsymbol{h}_{v_n,0} = \boldsymbol{s}_{v_n}$, based on its shallow feature $\boldsymbol{s}_{v_n}$. Then, the GNN adopts message-passing and aggregation patterns to learn structure-aware embedding for node $v_n$ via exploring the global context across all nodes as follows:

$$\boldsymbol{h}_{v_n,k} = \text{UPD}\left(\boldsymbol{h}_{v_n,k-1}, \text{AGG}\left(\{\text{MSG}\left(\boldsymbol{h}_{v_n,k-1}, \boldsymbol{h}_{v_m,k-1}\right)\}_{v_m \in \mathcal{N}(v_n)}\right)\right), \quad (1)$$

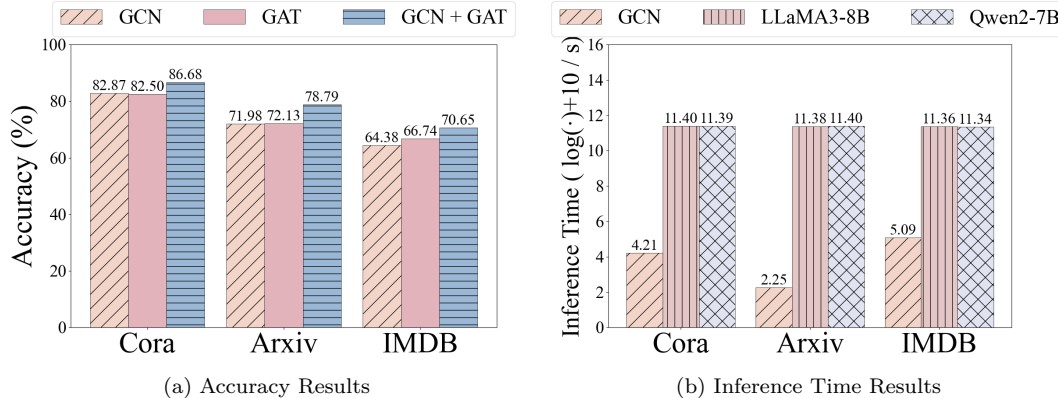

Figure 3: (a) Accuracy of GCN and GAT on validations of three datasets. "GCN + GAT" represents the accuracy on samples where both models make consistent predictions. (b) Inference efficiency comparison among GCN, LLaMA3-8B, and Qwen2-7B.

where $\boldsymbol{h}_{v_n,k}$ denotes the embedding of node $v_n$ in the $k$-th layer of the GNN and $\mathcal{N}(v_n)$ denotes the neighbor nodes set of $v_n$. The MSG function receives from each neighbor node's message, the AGG function is used for aggregating the neighbor node embeddings, and the UPD function updates the embedding of $v_n$ based on the aggregated neighbor node embeddings.

After obtaining the last layer's node embedding $\boldsymbol{h}_{v_n,K}$ via Eq. 1, we apply the predictor and the objective function tailored to the graph analytical task $\tau_i$ for training the GNN, where we take node classification $\tau_1$ as an example:

$$\hat{y}_{\tau_1,v_n} = \text{Softmax}\left(\text{MLP}_{\tau_1}\left(\boldsymbol{h}_{v_n,K}\right)\right), \tag{2}$$

$$\mathcal{L}_{\tau_1} = \mathbb{E}_{v_n \in \mathcal{V}} \text{CE}\left(\hat{y}_{\tau_1,v_n} | y_{\tau_1,v_n}\right), \tag{3}$$

where $\text{CE}(\cdot,\cdot)$ is the cross-entropy loss between the prediction $\hat{y}_{\tau_1,v_n}$ and the ground-truth label $y_{\tau_1,v_n}$ for node $v_n$. $K$ is the total number of GNN layers.

### 3.2.2 Conflict Evaluation via GNN-based Graph Agents Inference

Different GNN-based graph agents may generate inconsistent predictions for the same scenario on the TAG dataset, due to architectural differences. To fully integrate the opinions of various GNN experts, we propose a conflict evaluation mechanism. Specifically, for each input instance (e.g., a node in classification), if all GNN experts produce the same inference result, the result is directly accepted. Otherwise, the instance is treated as a conflict scenario and passed to LLMs for collaborative reasoning with the corresponding GNN outputs (cf., Section 3.4). This output-level detection is model-agnostic, enabling straightforward extension to settings with more experts, heterogeneous architectures, or different domains, while avoiding reliance on any single model's bias. We verify the impact of different GNN expert combinations and quantify the corresponding proportion of conflict scenarios in Section 4.3. Particularly, we automatically textualize each GNN expert's characteristic and inference result on the conflict scenario via a simplified template as follows:

> **A simplified opinion template for GNN expert $GNN_g$**
>
> **GNN Role:** $GNN_g$ is a graph analysis expert, depending on ... to form the node representations.
> **Answer:** {For node classification $\tau_1$, "Category Name".}

This mechanism balances effectiveness with computational efficiency based on multiple GNN-based graph agents, reducing the number of scenarios requiring further multi-agent collaboration. Meanwhile, it also ensures that each GNN expert's opinion can interact with other agents in the multi-agent framework. Section 4.6 provides the detailed efficiency analysis of `GMAgent`.

### 3.3 Repurposing LLMs as Graph Agents

TAGs with abundant attributes and flexible structures pose significant challenges for LLMs (Tang et al., 2024a; Ye et al., 2024), where LLMs as graph agents may struggle to precisely understand these unfamiliar graph structures and process new graph analytical tasks. This leads to potential inaccuracies and limited reasoning capabilities on various datasets. Instruction tuning enables LLMs to understand and perform well a wide range of graph analytical tasks on multiple datasets and contexts (Wang et al., 2023b;c). The effectiveness of such tuning for LLMs largely depends on how the instructions are structured. However, manually constructing these task-specific instructions is often time-consuming and requires excessive resources.

To address these challenges, we repurpose LLMs as graph agents. We first generate an LLM-powered graph agent via graph-driven instruction tuning, where we integrate CoT-based instructions from advanced LLMs (e.g., GPT-4o (Achiam et al., 2023)) with task-specific instructions (e.g., node classification) as the training corpus. Additionally, we employ the role-play expert recruiting strategy to gather diverse LLM graph experts into the multi-agent framework, and then obtain their initial analyses of conflict scenarios (c.f., Section 3.2.2).

#### 3.3.1 Generating LLM-powered Graph Agent

To help LLMs understand complex TAGs and execute graph analytical tasks, it is crucial to construct an effective training corpus specific to TAG data. Inspired by traditional graph analysis techniques, such as neighbors (Kipf & Welling, 2017; Hamilton et al., 2017; Yang et al., 2020b) and random walks (Li et al., 2021b; Ivanov & Burnaev, 2018; Tan et al., 2023), we propose an effective graph description textualization mechanism to describe graphs via these concepts. These techniques play complementary roles in understanding TAGs, providing LLMs a comprehensive and enriched perspective on graph structures (Tan et al., 2025; Fang et al., 2025). Particularly, neighbors facilitate the understanding of local connectivity and feature distributions, offering fine-grained insights into node attributes. In contrast, random walks serve as a dynamic strategy to capture global structural patterns and high-order relationships, highlighting the diversity of connectivity and paths. Specifically, we extract the key information from TAGs and convert it into a textual graph description for each node, consisting of multiple one-hop neighbors and three random walks. A simplified example of an effective graph description is given below:

> **A simplified effective graph description example**
>
> The effective graph description of $v_1$ is listed as follows:
> **Ego Graph Node:** $\{v_1: v_1\text{'s text attribute } x_1, v_2: v_2\text{'s } \dots\}$
> **One-hop Neighbor:** $\{v_2, v_3, v_4, v_5\}$
> **Random Walk:** $\{(1)\ v_1 \rightarrow v_2 \rightarrow v_6 \rightarrow v_7 \rightarrow v_8; \dots\}$

Based on this graph description, we then construct the task-specific instructions (e.g., node classification). To further improve LLM reasoning capabilities on unfamiliar graph structures or new tasks, we leverage the Chain-of-Thought (CoT) methodology (Wei et al., 2022). This allows GPT-4o to reason step-by-step based on the graph descriptions for different graph analytical tasks, generating answers accordingly. We then integrate these outputs from GPT-4o into CoT-based instructions for fine-tuning LLMs. The detailed prompt templates are provided in Appendix A.7.

For fine-tuning, we adopt a general LLM Qwen2-7B with LoRA (Hu et al., 2022) as the starting point, which can be flexibly replaced with other powerful LLMs. Then, we utilize the negative log-likelihood loss as the fine-tuning objective as follows:

$$p_\theta\left(Y_{j,k}|I_j, Y_{j,<k}\right) = LLM_\theta\left(I_j, Y_{j,<k}\right), \tag{4}$$

$$\mathcal{L}_{FT} = -\sum_{k=1}^{|Y_j|} \log p_\theta\left(Y_{j,k}|I_j, Y_{j,<k}\right), \tag{5}$$

where $\theta$ is the learnable parameters of the LLM, the instruction $I_j \in \mathcal{I}$ is the input of LLM, and $Y_j$ is the output of LLM. After obtaining the fine-tuned LLM for TAG analysis, we can repurpose it as the graph agent

for accurately analyzing and executing graph analytical tasks. Appendix A.5 further evaluates the impact of various graph description strategies for LLM-based graph agents.

### 3.3.2 Graph Expert Recruiting

LLMs often produce similar answers when given the same prompts due to their homogeneity (Lu et al., 2024; Padmakumar & He, 2024). This hinders discussions and decision-making in the multi-agent framework. To mitigate this issue, we introduce the role-play expert recruiting strategy. Here, we assign distinct roles to the graph agent from Section 3.3.1 as LLM graph experts. These roles focus on different perspectives, such as analyzing random walks or one-hop neighbors. In this way, we gather diverse analyses of conflict scenarios identified in Section 3.2.2. Specifically, given a conflict scenario $P_{c_i}$ and a role prompt $R_{LLM_l}$, we can generate the analysis from each LLM graph expert:

$$A_{LLM_l} = LLM_\theta \left( P_{c_i}, R_{LLM_l} \right). \tag{6}$$

We can obtain initial analyses from all LLM graph experts $\mathcal{A} = \{A_{LLM_1}, A_{LLM_2}, \ldots, A_{LLM_{M_L}}\}$, where $M_L$ is the total number of LLM graph experts. Then, we combine each analysis with the corresponding LLM expert's characteristics via the following template:

> **A simplified opinion template for LLM expert $LLM_l$**
>
> **LLM Role:** $LLM_l$ is a graph analysis expert, specializing in . . . . Its task is to analyze the graph based on . . . .
> **Answer:** {For node classification $\tau_1$, "Category Name".}
> **Analysis:** {$A_{LLM_l}$.}

### 3.4 Graph-oriented Multi-agent Collaboration

With LLMs demonstrating strong abilities in task understanding and self-planning, an emerging research direction is to build LLM-based multi-agent systems for graph analytical tasks (Ren et al., 2024; Hu et al., 2024; Wang et al., 2023a). Recently, GraphAgent-Reasoner (Hu et al., 2024) applied LLM-based collaboration for graph reasoning, handling graphs with over 1,000 nodes. However, how to integrate both GNN-based graph agents and LLM-based graph agents into the multi-agent collaboration for effectively and efficiently handling graph analytical tasks (e.g., node classification) remains unknown.

To address the issue, we propose a novel graph-oriented multi-agent collaboration method that fully leverages the strengths of GNN and LLM experts. This allows flexible interactions between diverse GNN-based and LLM-based graph agents, enhancing the accuracy of graph analytical tasks. Specifically, we propose to utilize advanced LLMs (e.g., GPT-4o) to assign a confidence score for each LLM expert and generate a summary report. This report, along with all experts' analyses from GNN experts and LLM experts, guides collaborative self-reflection among LLM experts and facilitates the final answer selection.

### 3.4.1 Summary Report Generation

Different LLM experts may offer varying answers based on their analytical perspectives (e.g., from random walks or one-hop neighbors). LLM-based graph agents often struggle to detect and correct their own mistakes, leading to potential misguidance during multi-agent collaboration (Chih-Yao Chen et al., 2025; Wang et al., 2024a). To this end, we utilize GPT-4o to assign a confidence score from 1 to 5 for each LLM expert based on their current analyses and generate a summary report by extracting key insights and a global summary (shown in Figure 2). With billions of parameters, GPT-4o, efficiently summarizes LLM expert insights (Tang et al., 2024b; Yeh et al., 2024). This summary guides LLM graph agents in prioritizing their considerations based on the reliability of each expert's analysis. Since GNN experts focus on structure but lack interpretability, GPT-4o only processes LLM analyses for scoring and summarization.

### 3.4.2 Collaborative Self-reflection Optimization

After generating the summary report, LLM experts iteratively refine their analyses through self-reflection, combining insights from other experts and the report. To measure the degree of agreement among the experts, we define the agreement across expert predictions based on the concept of entropy (Shannon, 1948) as follows:

$$\mathbb{CONS}_{T_i} = -\sum_{j=1}^{N} \mathbb{FRE}_{T_i,a_j} \cdot \log(\mathbb{FRE}_{T_i,a_j}), \tag{7}$$

where $N$ is the total number of candidate answers provided by experts and $\mathbb{FRE}_{T_i,a_j}$ is the frequency of each candidate answer $a_j$ appearing across experts' analyses at the current iteration $T_i$. In each iteration, LLM experts reflect on their previous analyses, where they also consider the summary report and insights from other experts to decide whether to revise their answers and analyses. We then leverage GPT-4o to generate a new summary report based on experts' revised analyses for the next iteration. Note that, this collaboration cycle is repeated until either $\mathbb{CONS}_{T_i} > \mathbb{CONS}_{T_{i-1}}$, or the maximum iteration count $T_{max}$ is reached.

Finally, we compute the final score for each predicted answer based on its frequency (c.f., Eq. 7) and each expert's confidence score $\mathbb{CONF}_{T_c,k}$, selecting the highest-scoring answer as final answer:

$$\hat{a}_{final} = \arg\max \sum_{j=1}^{N} \sum_{k=1}^{M} \mathbb{CONF}_{T_c,k} \cdot \mathbb{FRE}_{T_c,a_j}, \tag{8}$$

where $M$ is the total number of all experts, consisting of $M_G$ GNN-based graph agents and $M_L$ LLM-based graph agents. $T_c$ is the iteration at which the collaboration cycle concludes. Each LLM expert's confidence score is derived from the summary report at the iteration $T_c$. Notably, since GNN-based graph agents capture the global and structural information across the whole TAG, they are always assigned the highest confidence score of 5 by default. This graph-oriented multi-agent collaboration flexibly integrates the global and local insights from both GNN-based and LLM-based experts, effectively resolving conflicting scenarios. Furthermore, employing $\mathbb{CONS}_{T_i}$, as a criterion for determining when collaboration should stop, improves the efficiency of multi-agent collaboration. Appendix A.7 provides the prompt templates used in graph-oriented multi-agent collaboration.

### 3.5 Complexity Analysis

We analyze the time complexity of the proposed `GMAgent` framework by three major components: Deploying GNNs as Graph Agents, Repurposing LLMs as Graph Agents, and Graph-oriented Multi-agent Collaboration.

- *Deploying GNNs as Graph Agents.* For each GNN agent, the time complexity per layer is $\mathcal{O}(|\mathcal{V}|d + |\mathcal{E}|d)$, where $|\mathcal{V}|$ and $|\mathcal{E}|$ denote the number of nodes and edges in the graph, and $d$ is the dimension of node feature. With $K$ layers and $M_G$ GNN agents, the total cost is $\mathcal{O}(M_G \cdot K \cdot (|\mathcal{V}| + |\mathcal{E}|)d)$, which scales linearly with graph size and is efficient due to the parallelizable message passing.

- *Repurposing LLMs as Graph Agents.* Given $M_L$ LLM agents, each inference time on a textualized graph description is $\mathcal{O}(T_L \cdot L')$, where $T_L$ is the LLM's generation steps and $L'$ is the input prompt length. Instruction tuning is a one-time offline step with standard LLM fine-tuning complexity. Since LLMs process each conflict scenario independently, the per-scenario cost is $\mathcal{O}(M_L \cdot T_L \cdot L')$.

- *Graph-oriented Multi-agent Collaboration.* In each self-reflection iteration, all $M_L$ LLM experts revise their outputs based on the summary report derived from GPT-4o. GPT-4o performs scoring and summarization with cost $\mathcal{O}(T_S \cdot L'')$, where $T_S$ is the generation steps and $L''$ is the combined input length. Since the $M_L$ LLM experts generate responses independently, this step can be executed in parallel. Suppose the number of iterations is bounded by $T_{max}$, the total collaboration complexity per conflict scenario is $\mathcal{O}(T_{max} \cdot (M_L \cdot T_L \cdot L' + T_S \cdot L''))$.

In general, our proposed `GMAgent` balances accuracy and computational efficiency for graph analytical tasks, where the GNN-based agent deployment ensures low cost on large-scale graphs, while LLM-based agent

Table 1: Statistics of the used datasets.

| Task | Node Classification | | | | Link Prediction | |
|---|---|---|---|---|---|---|
| Dataset | Arxiv | Cora | IMDB | Products | PubMed | DBLP |
| # Nodes | 169,343 | 2,708 | 21,420 | 2,449,029 | 63,109 | 18,405 |
| # Edges | 1,166,243 | 5,429 | 86,642 | 61,859,140 | 244,986 | 67,946 |
| # Node Type | 1 | 1 | 4 | 1 | 4 | 3 |
| # Link Type | 1 | 1 | 6 | 1 | 10 | 4 |
| # Features | 128 | 1,433 | 3,489 | 100 | 200 | 334 |

collaboration is selectively applied only to conflict scenarios, reducing the frequency of LLM invocations. Additionally, we provide a detailed efficiency comparison of our `GMAgent` in Section 4.6.

## 4 Experiment

In this section, we evaluate our `GMAgent` framework, focusing on the following five key research questions:

- **RQ1:** How does our proposed `GMAgent` framework perform in comparison to the representative graph-oriented methods, and how well does it scale when applied to large-scale graphs?

- **RQ2:** How does `GMAgent` perform when integrating different GNN-based graph agents?

- **RQ3:** How is the performance of `GMAgent` affected by the choice of LLM-based graph agents?

- **RQ4:** How do different multi-agent collaboration strategies (e.g., number of agents, choices of summary agents, maximum iteration count, and self-reflection strategies) affect the performance of `GMAgent`?

- **RQ5:** How does `GMAgent` perform in terms of efficiency compared with the representative baselines?

### 4.1 Experimental Setup

#### 4.1.1 Datasets

To comprehensively evaluate the effectiveness and efficiency of our `GMAgent`, we utilize four real-world datasets for node classification (i.e., ogbn-arxiv (abbr. Arxiv[1]), Cora[2], IMDB[3], and ogbn-products (abbr. Products[1])), two for link prediction (i.e., PubMed[4] and DBLP[5]). The detailed statistics are shown in Table 1. We provide further details for the datasets in Appendix A.1.

#### 4.1.2 Evaluation Protocols

For node classification, we adopt different ratios of 54%/18%/28% for Arxiv, 60%/20%/20% for Cora, and 24%/6%/70% for IMDB, which is consistent with Hu et al. (2020a); He et al. (2024); Lv et al. (2021). Notably, we follow the official sales-ranking-based split from Hu et al. (2020a) for the large-scale Products dataset, where 8%/2%/90% of products are used for training, validation, and testing, respectively. For link prediction, we train all methods using the randomly selected 80% of links and evaluate them on the remaining 20% held-out links for PubMed, and employ the ratio of 80%/10%/10% for DBLP, following Yang et al. (2020a); Nguyen et al. (2023). We use two commonly adopted evaluation metrics for node classification (Yang et al., 2020a; Lv et al., 2021; Tan et al., 2023): Macro-F1 (across all labels) and Micro-F1 (across all nodes). The F1 score is a metric of the model's accuracy in binary and multi-class classification tasks, which considers both precision and recall. For link prediction, we compute the AUC and Accuracy (abbr. ACC) metrics as

---

[1]https://ogb.stanford.edu/
[2]http://www.cora.justresearch.com/lander
[3]https://www.kaggle.com/karrrimba/movie-metadatacsv
[4]https://www.ncbi.nlm.nih.gov/pubmed/
[5]https://dblp.uni-trier.de

suggested in Yang et al. (2020a); Tan et al. (2023); Liu et al. (2024a). AUC indicates the model's ability to distinguish between positive and negative classes across thresholds, while ACC represents the proportion of correctly classified instances overall.

### 4.1.3 Methods for Comparison

The following 14 characteristic baseline methods can be classified into two categories:

- **GNN-based method:** GCN (Kipf & Welling, 2017), GAT (Veličković et al., 2018), RevGNN (Li et al., 2021a), GraphSAGE (Hamilton et al., 2017), HGT (Hu et al., 2020b), HINormer (Mao et al., 2023), and TAPE (He et al., 2024).

- **LLM-based method:** Baichuan2-7B (Yang et al., 2023), Qwen2-7B (Yang et al., 2024a), LLaMA3-8B (Dubey et al., 2024), GPT-3.5 Ouyang et al. (2022), GPT-4 (Achiam et al., 2023), GraphGPT (Tang et al., 2024a), and MARK (Fu et al., 2025).

For more details of the compared baselines, please refer to Appendix A.2.

### 4.1.4 Implementation Details

For our `GMAgent`, we utilize AutoGen[6] and FastChat[7] to enable collaboration among multiple agents. By default, we select fine-tuned Qwen2-7B as the foundation model for our LLM-based graph agent and employ GCN and GAT for our GNN-based graph agents. The number of LLM-based graph agents $M_L$ is set to 4, and the maximum iteration count $T_{max}$ is set to 2. Qwen2-7B is fine-tuned using LLaMA-Factory (Zheng et al., 2024) on a mixture of instructions from a wide range of tasks and datasets. We perform parameter-efficient fine-tuning via LoRA (Hu et al., 2022) with rank $r = 32$ and scaling factor $\alpha = 64$. We set the learning rate to $5 \times 10^{-5}$, the maximum input length of the LLM to 1200 tokens, and train for two epochs. GNN-based methods are trained and evaluated using CogDL (Cen et al., 2023) or HGB (Lv et al., 2021). For LLM-based methods, we load the checkpoint of LLM from HuggingFace[8] or call the official API from OpenAI[9] for evaluation. All LLM-based methods and `GMAgent` are evaluated using the same test instructions. All experiments are conducted using eight NVIDIA GTX 3090 Ti GPUs. Notably, due to the reliance on manually customized prompts and GPT-3.5 APIs to generate high-quality explanations for each dataset, we only use publicly available TAPE features on the Cora and Arxiv datasets[10]. Therefore, the results for TAPE (He et al., 2024) are excluded from Table 2. However, we did incorporate TAPE into the GNN graph agents variation comparison in Section 4.3 on the Arxiv and Cora datasets, using feature files sourced from the TAPE repository. The full code for this work is available[11].

### 4.2 Overall Performance (RQ1)

In this subsection, we provide a comprehensive performance analysis of our proposed `GMAgent` framework across various graph analytical tasks and datasets, comparing it with the state-of-the-art baselines. This evaluation focuses on both node classification and link prediction tasks, assessing the model's ability to understand and predict using Text-Attributed Graphs (TAGs).

Overall, our proposed `GMAgent` demonstrates superior performance, attributed to its unique integration of GNNs for capturing global information and LLMs for interpreting textual attributes. As shown in Table 2, Table 3, and Figure 4, our `GMAgent` consistently surpasses all baseline models in every evaluation metric. By efficiently assigning simpler tasks to GNNs and utilizing LLMs to handle complex, text-heavy scenarios, `GMAgent` achieves optimal resource allocation and high accuracy across six datasets. In addition, the comprehensive performance analyses of `GMAgent` on link prediction are presented in Appendix A.3.

---

[6]https://github.com/microsoft/autogen
[7]https://github.com/lm-sys/FastChat
[8]https://huggingface.co
[9]https://platform.openai.com
[10]https://github.com/XiaoxinHe/TAPE
[11]https://github.com/lvhangkenn/GMAgent

Table 2: Experimental results (%) on three datasets for node classification, where * denotes a significant improvement according to the Wilcoxon signed-rank test (Woolson, 2007). The best performances are highlighted in **boldface** and the second runners are underlined.

| Dataset | Arxiv | | Cora | | IMDB | |
|---|---|---|---|---|---|---|
| Method | Micro-F1 | Macro-F1 | Micro-F1 | Macro-F1 | Micro-F1 | Macro-F1 |
| GCN | $71.73_{\pm0.24}$ | $51.12_{\pm0.65}$ | $81.87_{\pm0.55}$ | $81.29_{\pm0.76}$ | $64.35_{\pm0.73}$ | $58.17_{\pm1.45}$ |
| GAT | $72.24_{\pm0.31}$ | $\underline{52.30}_{\pm0.54}$ | $81.50_{\pm0.68}$ | $81.42_{\pm0.53}$ | $64.31_{\pm0.94}$ | $58.94_{\pm1.30}$ |
| RevGNN | $\underline{72.76}_{\pm0.28}$ | $51.38_{\pm0.62}$ | $84.19_{\pm0.42}$ | $82.21_{\pm0.90}$ | $65.89_{\pm0.81}$ | $59.90_{\pm1.21}$ |
| GraphSAGE | $71.45_{\pm0.57}$ | $50.75_{\pm0.86}$ | $\underline{84.53}_{\pm0.36}$ | $\underline{83.68}_{\pm0.41}$ | $62.34_{\pm0.79}$ | $53.57_{\pm1.96}$ |
| HGT | $71.25_{\pm0.52}$ | $51.39_{\pm0.75}$ | $82.61_{\pm0.31}$ | $81.05_{\pm0.64}$ | $67.12_{\pm0.65}$ | $63.38_{\pm1.57}$ |
| HINormer | $71.08_{\pm0.49}$ | $51.77_{\pm0.81}$ | $82.84_{\pm0.57}$ | $81.28_{\pm0.87}$ | $\underline{67.47}_{\pm0.58}$ | $\underline{64.09}_{\pm1.09}$ |
| Baichuan2-7B | $2.43_{\pm1.23}$ | $1.98_{\pm1.09}$ | $8.90_{\pm2.12}$ | $5.04_{\pm2.55}$ | $40.55_{\pm2.41}$ | $39.14_{\pm2.06}$ |
| Qwen2-7B | $40.25_{\pm2.84}$ | $18.56_{\pm2.41}$ | $58.54_{\pm1.35}$ | $48.35_{\pm1.83}$ | $64.48_{\pm2.69}$ | $61.27_{\pm2.28}$ |
| LLaMA3-8B | $23.16_{\pm2.57}$ | $11.26_{\pm2.15}$ | $14.76_{\pm2.06}$ | $8.49_{\pm1.98}$ | $37.65_{\pm1.90}$ | $34.86_{\pm2.83}$ |
| GPT-3.5 | $43.23_{\pm2.30}$ | $32.28_{\pm2.59}$ | $65.30_{\pm1.14}$ | $55.34_{\pm1.52}$ | $55.49_{\pm1.47}$ | $54.14_{\pm2.03}$ |
| GPT-4 | $51.21_{\pm1.95}$ | $43.40_{\pm2.07}$ | $67.72_{\pm1.89}$ | $56.14_{\pm1.35}$ | $59.57_{\pm1.19}$ | $58.18_{\pm2.55}$ |
| GraphGPT | $31.48_{\pm1.52}$ | $17.62_{\pm2.30}$ | $24.47_{\pm2.06}$ | $15.16_{\pm2.41}$ | $44.25_{\pm1.58}$ | $43.51_{\pm2.69}$ |
| MARK | $55.42_{\pm2.60}$ | $44.70_{\pm2.59}$ | $72.35_{\pm2.13}$ | $69.32_{\pm2.67}$ | — | — |
| GMAgent | $\mathbf{78.72^*}_{\pm0.91}$ | $\mathbf{59.30^*}_{\pm0.85}$ | $\mathbf{85.97^*}_{\pm0.89}$ | $\mathbf{85.61^*}_{\pm1.02}$ | $\mathbf{74.36^*}_{\pm1.03}$ | $\mathbf{66.82^*}_{\pm1.24}$ |

Table 3: Experimental results (%) on the Products dataset for node classification, where * denotes a significant improvement according to the Wilcoxon signed-rank test (Woolson, 2007). Total runtime is measured on 50,000 randomly sampled test instances.

| Dataset | Products | | | | |
|---|---|---|---|---|---|
| Method | GCN | GAT | Qwen2-7B | LLaMA3-8B | GMAgent |
| Micro-F1 | $76.38_{\pm0.26}$ | $73.18_{\pm0.38}$ | $58.02_{\pm2.30}$ | $44.23_{\pm2.53}$ | $\mathbf{81.65^*}_{\pm1.12}$ |
| Macro-F1 | $38.07_{\pm0.69}$ | $35.06_{\pm0.61}$ | $38.81_{\pm2.15}$ | $32.57_{\pm2.69}$ | $\mathbf{44.27^*}_{\pm1.08}$ |
| Total Runtime | 4min49s | 23min25s | 340h | 367h | 81h |
| Inference Memory Usage | 15.3GB | 23.1GB | 15.8GB | 16.6GB | 16.7GB |

### 4.2.1 Comparison with GNN-based Methods

Generally, our proposed `GMAgent` consistently outperforms GNN-based methods across all tasks and datasets, showcasing its precise understanding of graph data (shown in Table 2, Table 3, and Figure 4). `GMAgent` achieves significant performance gains in node classification and link prediction with an average of 7.63% and 3.58%, respectively. Notably, our framework achieves an average improvement of 11.84% over standard GCN and GAT metrics. While approaches like RevGNN excel in node classification and HGT in link prediction, both struggle to fully utilize the rich attributed text in TAGs for challenging scenarios. In contrast, `GMAgent` capitalizes on the LLMs' ability to understand semantic content, boosting performance in challenging scenarios where traditional GNN-based models fall short.

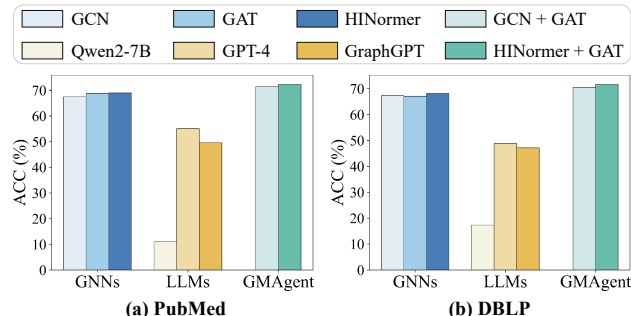

Figure 4: ACC results (%) on the PubMed and DBLP datasets for link prediction.

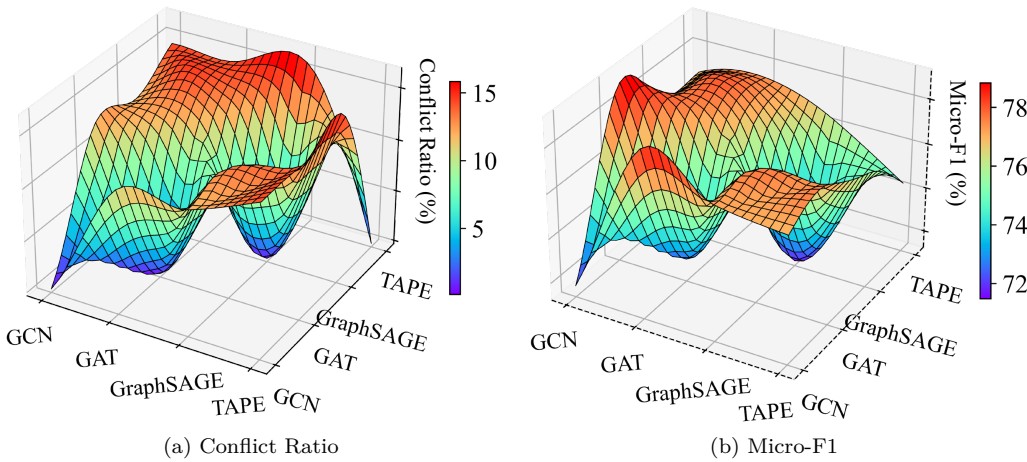

Figure 5: Influence of developing different GNNs as graph agents in our `GMAgent` on the Arxiv dataset.

### 4.2.2 Comparison with LLM-based Methods

As illustrated in Table 2, Table 3, and Figure 4, `GMAgent` also surpasses all LLM-based methods with significant improvements on various tasks and datasets, highlighting `GMAgent`'s superior capabilities in TAG analysis. Compared to Qwen2-7B, `GMAgent` achieves an overall average improvement of 179.20% in node classification and link prediction. Despite GPT-4o's robust generalization abilities based on extensive parameter sets, it faces difficulties when dealing with complex graph structures and struggles with fine-tuning on unfamiliar datasets. By considering both efficiency and cost, `GMAgent` utilizes fine-tuned Qwen and LLaMA series models, yet surpasses GPT-4o with both GNNs' global structural insights and the LLMs' semantic understanding. Notably, MARK relies on graph clustering to identify uncertain nodes for LLM-based multi-agent collaboration, which limits its applicability to tasks such as multi-label node classification (e.g., IMDB) or link prediction (e.g., PubMed). Therefore, we mark "—" in Table 2 for IMDB. In contrast, our `GMAgent` offers a flexible multi-agent collaboration framework that effectively integrates the global structural learning abilities of GNNs and the local semantic richness of LLMs, enabling broad adaptability across diverse graph analytical tasks.

### 4.2.3 Evaluation on Large-scale Graphs

To assess the scalability and resource demands of `GMAgent` on large-scale graphs, we conduct experiments on Products, which is an undirected and unweighted Amazon co-purchasing network with 2.45M nodes and 61.86M edges. As shown in Table 3, `GMAgent` achieves the best Micro-F1 and Macro-F1, while maintaining reasonable runtime (81h) and inference memory usage (16.7GB). Compared to LLM-based baselines such as Qwen2-7B (340h, 15.8GB) and LLaMA3-8B (367h, 16.6GB), `GMAgent` efficiently reduces total runtime by over 75% and significantly improves overall performance (achieving up to an 84.60% gain in Micro-F1 over LLaMA3-8B). Moreover, its memory usage remains comparable to GNN-based methods (e.g., 16.7GB for `GMAgent` vs. 23.1GB for GAT). These results demonstrate that `GMAgent` scales well even on million-node graphs, achieving a practical balance between predictive accuracy and computational efficiency. They also highlight its potential for practical deployment in real-world industrial applications.

### 4.3 Varying GNN-based Graph Agents (RQ2)

Figure 5 illustrates how different combinations of GNN-based graph agents (GCN, GAT, GraphSAGE, and TAPE (GCN)) affect the conflict ratio and Micro-F1 on the Arxiv dataset within `GMAgent`.

In general, we observe that integrating any GNN agent combinations leads to a certain proportion of conflict scenarios, with an average conflict ratio of 13.29% across all combinations and a range of 11.61%-15.96%. This indicates that the conflict evaluation mechanism (Section 3.2.2) filters most consistent predictions as final

Table 4: Influence of different LLM-based graph agents.

| Dataset | Arxiv | | Cora | |
|---|---|---|---|---|
| Metric | Micro-F1 | Macro-F1 | Micro-F1 | Macro-F1 |
| LLaMA3-8B | 73.84 | 54.68 | 80.51 | 79.64 |
| Qwen2-7B | 75.53 | 55.09 | 81.29 | 80.92 |
| LLaMA3-8B$_{FT}$ | 77.19 | 58.45 | 83.47 | 82.85 |
| Qwen2-7B$_{FT}$ | **78.72** | **59.30** | **85.97** | **85.61** |

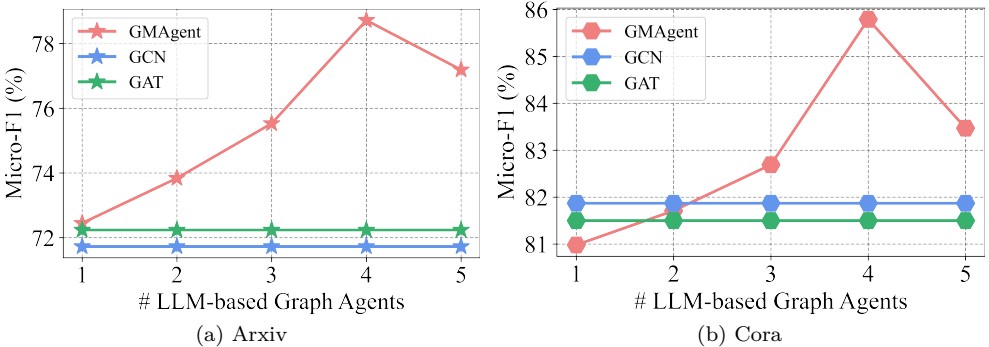

(a) Arxiv      (b) Cora

Figure 6: Influence of the number of LLM-based graph agents.

outputs, alleviating the frequent call to costly LLMs. Resolving the remaining conflicts via LLM collaboration yields potential improvements in predictive accuracy, with an average Micro-F1 improvement of 6.70% over the corresponding single-GNN baselines.

Furthermore, we observe that performance varies across different combinations of GNN agents. For example, GCN+GAT achieves the highest Micro-F1 with the lowest conflict ratio, which aligns with their complementary structural focuses: GCN models global topology and GAT emphasizes node-level attention. In contrast, combinations that involve TAPE (e.g., GCN+TAPE) exhibit higher conflict ratios but only marginal gains on Arxiv. A likely reason is the misalignment between TAPE's LLM-derived semantic embeddings and the structure-oriented outputs of other GNNs, which increases disagreement without consistently improving accuracy. These observations suggest that selecting structurally complementary yet inference-aligned agents (e.g., GCN+GAT) is more effective, while semantically divergent agents should be introduced selectively and evaluated per dataset. Additional results on Cora are provided in Appendix A.4.

### 4.4 Varying LLM-based Graph Agents (RQ3)

Table 4 shows the influence of different LLM-based graph agents on two datasets, where fine-tuning significantly improves performance across all models. Qwen2-7B$_{FT}$, in particular, consistently outperforms other models on both datasets, achieving the highest values for both Micro-F1 and Macro-F1 with an average 2.44% improvement over the second-best foundation model LLaMA3-8B$_{FT}$.

Interestingly, the fine-tuned LLaMA3-8B also shows considerable improvement compared to its base version, though it still lags behind Qwen2-7B$_{FT}$. These results suggest that fine-tuning plays a critical role in enhancing the performance of LLM-based agents in graph analytical tasks.

### 4.5 Impact of Varying Graph-oriented Multi-agent Collaboration Strategies (RQ4)

#### 4.5.1 Varying Number of Agents

As shown in Figure 6, increasing the number of LLM-based graph agents consistently improves performance across both Arxiv and Cora datasets. However, adding more agents requires additional prompt design and

Table 5: Influence of different summary agents.

| Dataset | Arxiv | | Cora | |
|---|---|---|---|---|
| Metric | Micro-F1 | Macro-F1 | Micro-F1 | Macro-F1 |
| LLaMA3-8B | 72.54 | 49.36 | 81.14 | 79.43 |
| Qwen2-7B | 74.80 | 52.89 | 83.86 | 81.52 |
| GPT-4o | **78.72** | **59.30** | **85.97** | **85.61** |

(a) Arxiv        (b) Cora

Figure 7: Influence of the maximum iteration count.

increases computational costs, as more interactions and coordination between agents are needed. To balance between accuracy and computational efficiency, an optimal choice of agent number is 4.

### 4.5.2 Varying Summary Agents

Table 5 illustrates the influence of different LLM-based summary agents on performance. GPT-4o consistently outperforms LLaMA3-8B and Qwen2-7B, particularly on the text-heavy Arxiv dataset. This can be attributed to GPT-4o's superior capability in handling complex language and generating detailed, coherent summaries. The role of the summary agent is crucial in our proposed `GMAgent` framework as it consolidates insights from multiple graph agents, guiding the collaborative process toward a consensus. By providing a global overview and prioritizing key insights, GPT-4o effectively enhances the decision-making process within the multi-agent collaboration, contributing to more accurate final predictions.

### 4.5.3 Varying Maximum Iteration Count

As shown in Figure 7, we observe that moderately increasing the maximum iteration count $T_{max}$ from 1 to 2 can improve accuracy, as additional self-reflection rounds enable LLM-based graph agents to better refine and consolidate their predictions. However, beyond $T_{max} = 2$, the performance improvement declines, while the inference time continues to increase. This indicates that excessive iterations may lead to redundant reasoning and reduce the effectiveness of prediction refinement. Therefore, we adopt $T_{max} = 2$ as a practical trade-off, balancing predictive accuracy and computational efficiency while ensuring sufficient iterative refinement.

### 4.5.4 Varying Self-reflection Strategies

To study the effectiveness of different strategies during the multi-agent collaboration process, we compare three variants of our proposed method in the self-reflection process. We study `GMAgent` as follows:

- `GMAgent` w/o. $\mathcal{R}_{Re}$ represents `GMAgent` without using the summary report in the self-reflection process.

- `GMAgent` w/o. $\mathcal{R}_{Ot}$ represents `GMAgent` without using other agents' analyses in the self-reflection process.

- `GMAgent` w/o. $\mathcal{R}_{Pr}$ represents `GMAgent` without using prior self-analysis in the self-reflection process.

Table 6: Influence of different self-reflection strategies.

| Dataset | Arxiv | | Cora | |
|---|---|---|---|---|
| Metric | Micro-F1 | Macro-F1 | Micro-F1 | Macro-F1 |
| GMAgent | **78.72** | **59.30** | **85.97** | **85.61** |
| w/o. $\mathcal{R}_{Re}$ | 70.03 | 49.47 | 79.14 | 78.65 |
| w/o. $\mathcal{R}_{Ot}$ | 74.54 | 54.91 | 83.05 | 81.89 |
| w/o. $\mathcal{R}_{Pr}$ | 76.81 | 57.25 | 84.62 | 83.24 |

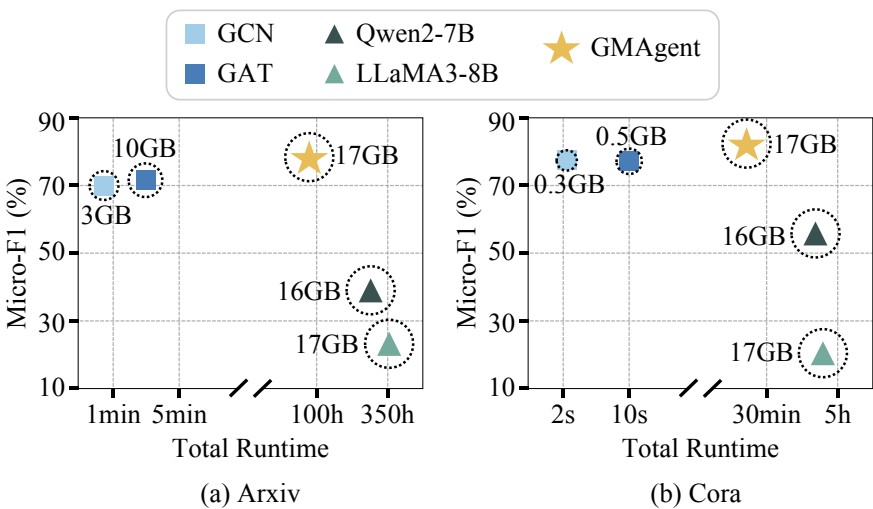

Figure 8: Comparison of different methods on Arxiv and Cora in terms of Micro-F1 (y-axis), total runtime (x-axis), and memory usage (dotted circles). Shapes denote method types (□ GNNs, △ LLMs, ☆ GMAgent).

As shown in Table 6, removing the summary report (GMAgent w/o. $\mathcal{R}_{Re}$) causes the most significant drop in performance, emphasizing the crucial role the report plays in guiding the self-reflection process. The summary report prompts agents to reconsider their earlier analyses based on consolidated feedback, facilitating deeper reflection and improvement in subsequent iterations. Without the summary report, the graph agents lack a clear direction for refinement, resulting in less effective self-reflection. Moreover, removing the analyses from other agents (GMAgent w/o. $\mathcal{R}_{Ot}$) or ignoring prior self-analysis (GMAgent w/o. $\mathcal{R}_{Pr}$) also leads to performance degradation, further underscoring the importance of collaborative feedback and the iterative self-reflection mechanism within the multi-agent framework. We provide a simplified scenario to illustrate the effectiveness of our collaborative self-reflection mechanism in Appendix A.6.

## 4.6 Efficiency Analysis (RQ5)

As shown in Figure 8, GMAgent consistently achieves a favorable trade-off between predictive accuracy and computational efficiency across both datasets. Compared with LLM-based baselines (e.g., Qwen2-7B), GMAgent reduces total runtime by 74.85% on Arxiv and 90.27% on Cora, while achieving the best predictive accuracy and maintaining comparable memory usage. This substantial efficiency gain stems from our conflict scenario evaluation mechanism (cf., Section 3.2.2), where the majority of instances can be accurately handled by strong GNN experts without invoking expensive LLM-based inferences. Additionally, despite a moderate increase in runtime and memory usage, GMAgent significantly outperforms GNN-based baselines (e.g., GCN and GAT) in predictive performance and provides superior interpretability and transparent reasoning via its multi-agent collaboration mechanism.

## 5 Conclusion and Future Work

In this paper, we introduce `GMAgent`, an effective and flexible graph-oriented multi-agent collaboration framework for text-attributed graph analysis. By enabling seamless interactions between diverse GNN-based and LLM-based graph agents, `GMAgent` integrates the global structural learning power of GNNs and the local semantic richness of LLMs to enhance graph analytical tasks. Specifically, `GMAgent` includes innovative deploying GNNs as graph agents, repurposing LLMs as graph agents, and enabling graph-oriented multi-agent collaboration among these graph agents. Extensive experiments on five datasets demonstrate `GMAgent`'s superior performance over the state-of-the-art baselines, improving not only the comprehension of graph data but also the accuracy and interpretability of graph analytical tasks.

Looking ahead, future work could improve `GMAgent`'s capabilities to tackle complex and conflicting scenarios in graph analysis. This includes creating better selection mechanisms (e.g., learned selectors or threshold-based confidence aggregation) to identify conflicting outputs, designing more effective graph-to-text transformation strategies to balance redundancy and information preservation, and adding a wider range of agents for decision-making. Moreover, it is also interesting to explore real-time agent collaboration strategies, where agents adjust roles based on problem complexity, thereby enhancing efficiency and adaptability.

### Acknowledgments

This work was supported in part by the Fujian Provincial Artificial Intelligence Industry Development Technology Project under Grant (2025H0042), Fujian Provincial Natural Science Foundation of China under Grants (2025J01540), National Natural Science Foundation of China under Grants (No.62302098). Carl Yang was not supported by any fund from China.

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

# A Appendix

## A.1 Detailed Descriptions of Datasets

This section provides detailed descriptions of each graph dataset used in our experiment.

(1) **Node classification**: Node classification assigns the target node to predefined categories by utilizing the diverse relationships and attributes present in the graph.

- ogbn-arxiv (abbr. Arxiv) represents a directed graph that captures the citation network among computer science arXiv papers indexed by MAG (Wang et al., 2020). Each paper in the dataset is associated with a research category, manually labeled by the authors and arXiv moderators. These research categories are selected from a set of 40 subject areas.

- Cora comprises 2,708 scientific publications classified into one of seven classes–case-based, genetic algorithms, neural networks, probabilistic methods, reinforcement learning, rule learning, and theory, with a citation network consisting of 5,429 links.

- IMDB is a website about movies and related information, including a subset from the Action, Comedy, Drama, Romance, and Thriller genres. Each labeled movie has one or multiple labels.

- ogbn-products (abbr. Products) is an undirected and unweighted Amazon co-purchasing network with 2.45M nodes and 61.86M edges, where each product is assigned to one of 47 categories for a multi-class node classification task.

(2) **Link prediction**: Link prediction predicts the likelihood of a future or missing connection between two nodes in a graph.

- PubMed contains a graph of genes, diseases, chemicals, and species. It performs word2vec computations on all PubMed papers and aggregates the word embeddings to generate 200-dimensional features for each type of node.

- DBLP includes a substantial collection of papers on the web, authors, conferences, and terms, providing a comprehensive dataset. The target nodes, representing authors, are categorized into four research areas: database, data mining, machine learning, and information retrieval.

## A.2 Detailed Descriptions of Baselines

The following characteristic baseline methods can be classified into two categories: (1) GNN-based methods and (2) LLM-based methods.

(1) **GNN-based methods**:

- GCN (Kipf & Welling, 2017) scales linearly in the number of graph edges and learns hidden layer representations that encode both local graph structure and features of nodes.

- GAT (Veličković et al., 2018) utilizes masked self-attention mechanisms to enhance the processing of graph data by addressing limitations in traditional graph convolution methods.

- RevGNN (Li et al., 2021a) captures long-range interactions in graph data and reduces memory complexity with grouped reversible connections, enabling more effective training of deep and wide GNNs.

- GraphSAGE (Hamilton et al., 2017) generates node embeddings by sampling and aggregating features from a node's local neighborhood, enabling scalable learning on large graphs.

- HGT (Hu et al., 2020b) extends the transformer architecture to handle heterogeneous graphs to capture diverse node and edge interactions.

- HINormer (Mao et al., 2023) uses graph transformers to learn node representations on heterogeneous information networks by capturing both local structure and heterogeneity.

- TAPE (He et al., 2024) leverages LLMs' explanations to generate informative node features for text-attributed graphs, boosting the performance of various GNNs.

(2) **LLM-based methods**:

- Baichuan2-7B-Base (abbr. Baichuan2-7B) (Yang et al., 2023) is an open-source, bilingual language model developed by Baichuan Inc., trained on 2.6 trillion tokens with 7 billion parameters.

- Qwen2-7B-Instruct (abbr. Qwen2-7B) (Yang et al., 2024a) is an instruction-tuned 7 billion parameter model, designed to excel in tasks like language understanding, generation, and more, with support for processing up to 131,072 tokens in context.

- LLaMA3-8B (Dubey et al., 2024) succeeds LLaMA2, offering improved performance with 8 billion parameters through advancements in architecture, training data, and optimization.

- GPT-3.5 (Ouyang et al., 2022) is a large-scale language model developed by OpenAI with 175 billion parameters, capable of generating human-like text and understanding complex contexts.

- GPT-4 (Achiam et al., 2023) builds upon GPT-3.5, providing advanced language generation and understanding capabilities with greater scale and improved performance.

- GraphGPT (Tang et al., 2024a) uses LLM as backbone and integrates LLMs with graph knowledge using a graph structural instruction tuning paradigm, enhancing understanding through text-graph grounding and step-by-step reasoning.

- MARK (Fu et al., 2025) is a multi-agent framework that enhances text-attributed graph clustering by leveraging three LLM-based agents to provide ranking-based supervision signals for refining uncertain nodes near cluster boundaries.

### A.3 Overall Performance on Link Prediction

In this subsection, we comprehensively analyze the performance of our proposed `GMAgent` framework on link prediction tasks across PubMed and DBLP datasets, comparing it with state-of-the-art baselines. Overall, our proposed `GMAgent` consistently demonstrates superior performance, which can be attributed to its effective integration of GNNs for capturing global structural information and LLMs for interpreting complex textual attributes. As shown in Figure 4 and Figure 9, our `GMAgent` outperforms all baseline models in both ACC and AUC metrics. By efficiently assigning simpler tasks to GNNs and utilizing LLMs to handle more complex, text-heavy scenarios, `GMAgent` achieves optimal resource allocation and high accuracy across both datasets.

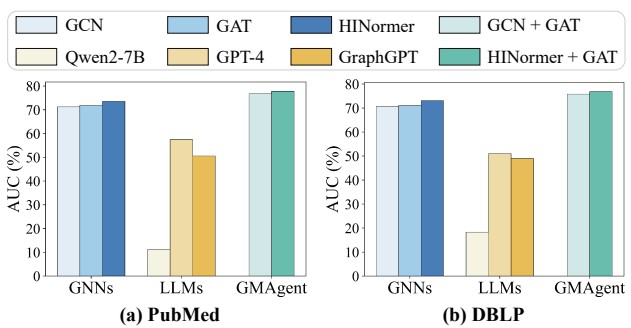

Figure 9: AUC results (%) on the PubMed and DBLP datasets for link prediction.

### A.3.1 Comparison with GNN-based Methods

Generally, our proposed `GMAgent` consistently surpasses GNN-based methods on link prediction tasks across PubMed and DBLP datasets, demonstrating its precise understanding of graph structures (shown in Figure 4 and Figure 9). Specifically, `GMAgent` (HINormer + GAT) yields a significant performance improvement, with an average gain of 5.22%. Moreover, `GMAgent` (GCN + GAT) and `GMAgent` (HINormer + GAT) achieve an average improvement of 6.03% and 6.18% over their corresponding standard models, respectively.

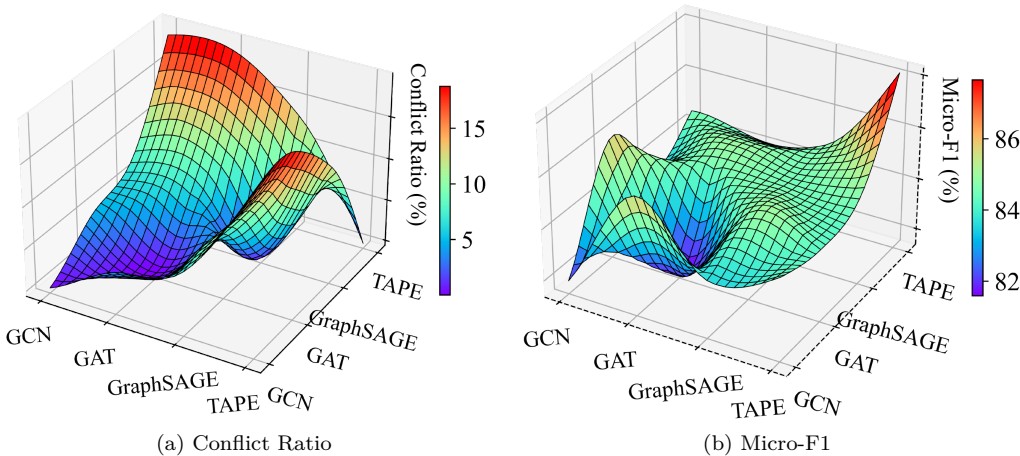

(a) Conflict Ratio                    (b) Micro-F1

Figure 10: Influence of developing different GNNs as graph agents in our `GMAgent` on the Cora dataset.

This showcases the accuracy of our graph-oriented multi-agent collaboration framework for TAG analysis. Traditional approaches, like HGT, struggle to fully utilize the rich text attributes of TAGs, particularly in more complex scenarios. In contrast, `GMAgent` fully harnesses the LLMs' ability to comprehend semantic content, boosting performance in challenging scenarios where the GNN-based models fall short.

### A.3.2   Comparison with LLM-based Methods

As illustrated in Figure 4 and Figure 9, `GMAgent` also outperforms all LLM-based methods with significant improvements on link prediction tasks across both datasets, highlighting `GMAgent` 's superior capabilities in TAG analysis. Compared to Qwen2-7B, `GMAgent` (HINormer + GAT) achieves an overall average improvement of 446.43% in link prediction, indicating the inherent limitation for general LLMs of understanding complex graph structures. Despite GPT-4o's robust generalization abilities based on extensive parameter sets, it faces difficulties when dealing with complex graph structures and struggles with fine-tuning on unfamiliar datasets. By considering both efficiency and cost, `GMAgent` utilizes fine-tuned Qwen and LLaMA models, yet surpasses GPT-4o with both GNNs' global structural insights and the LLMs' semantic understanding.

### A.4   Impact of Varying GNN-based Graph Agents on the Cora Dataset

Figure 10 illustrates how different combinations of GNN-based graph agents (GCN, GAT, GraphSAGE, and TAPE (GCN)) affect the proportion of conflict scenarios (i.e., conflict ratio) and Micro-F1 on the Arxiv dataset within `GMAgent`.

In general, we observe that integrating any GNN agent combinations leads to a certain proportion of conflict scenarios, with an average conflict ratio of 11.62% across all combinations and a range of 4.42%-19.00%. This indicates that the conflict evaluation mechanism (Section 3.2.2) filters most consistent predictions as final outputs, alleviating the frequent call to costly LLMs. Resolving the remaining conflicts via LLM collaboration yields potential improvements in predictive accuracy.

On Cora, performance also varies notably across GNN agent combinations. Single TAPE, which enriches a GCN backbone with semantic node attributes, achieves the highest standalone accuracy. This suggests that in datasets with relatively simple and homogeneous graph structures, a single strong GNN agent may already capture most of the task-relevant information, thereby performing accurate predictions. In such cases, incorporating additional agents can lead to redundant conflicts and even degrade overall performance. This finding reflects the dataset-specific advantage of TAPE rather than a limitation of `GMAgent`. On more complex datasets, like Arxiv, integrating TPAE with GCN/GAT with the help of our `GMAgent` consistently yields better performance than using TAPE alone (cf., Section 4.3). Overall, the conflict evaluation mechanism benefits from structurally complementary and inference-aligned pairings (e.g., GCN+GAT) on complex

Table 7: Influence of different graph description strategies.

| Dataset | Arxiv | | Cora | |
|---|---|---|---|---|
| Metric | Micro-F1 | Macro-F1 | Micro-F1 | Macro-F1 |
| Base | 75.53 | 55.09 | 81.29 | 80.92 |
| + Ego Graph Node | 76.21 | 56.85 | 82.34 | 82.01 |
| + One-hop Neighbour | 77.46 | 58.02 | 84.15 | 83.74 |
| GMAgent | **78.72** | **59.30** | **85.97** | **85.61** |

Figure 11: A node classification conflict scenario on Arxiv to illustrate the effectiveness of our collaborative self-reflection mechanism in GMAgent.

graphs, while semantically divergent agents (e.g., TAPE) require more selective integration based on graph structure and attribute richness.

## A.5 Impact of Different Graph Description Variants

We evaluate the effect of various graph description strategies for LLM-based agents on the Arxiv and Cora datasets as follows: (1) "Base" directly utilizes vanilla Qwen2-7B as graph agents without any graph-aware fine-tuning; (2) "+ Ego Graph Node" fine-tunes the LLM using only the central node attributes, providing basic node-level semantics; (3) "+ One-hop Neighbour" further integrates one-hop neighbors, enriching the local structural context; (4) GMAgent combines one-hop neighbors with random walks (cf., Section 3.3.1) to encode both local and global structural semantics, enabling a more comprehensive graph understanding. As shown in Table 7, model performance consistently improves with the integration of richer structural signals. Notably, our GMAgent setup achieves the best performance across all metrics on both datasets, demonstrating the effectiveness of combining local neighborhood information with global path-based context in the textualization process. The abundant textual graph descriptions provide diverse and complementary semantics, which enable LLMs to significantly comprehend and analyze graph structures.

## A.6 Case Studies

To assess the effectiveness of GMAgent's collaborative self-reflection mechanism in enhancing LLM graph agents' understanding of graph data and executing graph analytical tasks, we provide a node classification conflict scenario on the Arxiv dataset. Figure 11 shows the distinct responses of the One-Hop Neighbors Expert and Centrality Expert at different rounds of collaborative self-reflection, and the summary report generated by GPT-4o. In the first iteration, the One-Hop Neighbors Expert tends to predict cs.DS, influenced by neighboring nodes associated with "*multi-resolution hashing for fast pairwise summations*". In contrast,

the Centrality Expert, which focuses on centrality metrics (e.g., degree and closeness), suggests that cs.SI is a more appropriate category. Nevertheless, accurately identifying the ground-truth label (i.e., cs.DS) remains challenging due to the existing conflict scenario within different agent analyses.

Additionally, GPT-4o assigns a confidence score from 1 (Poor Confidence) to 5 (Strong Confidence) for each LLM expert based on their analysis (for instance, assigning a score of 5 to the One-Hop Neighbors Expert). Acting as a summary agent, GPT-4o extracts key insights from the various LLM graph agents' analyses, providing a global overview tailored to this conflict scenario. By integrating this summary report with the GNN graph agents' answer candidates, our `GMAgent` stabilizes the answer distribution of all graph agents during the collaborative self-reflection phase, thereby improving the prioritization of the ground-truth label. Particularly, guided by the summary report and insights from other agents, the Centrality Expert is able to identify the correct answer. These results strongly support the effectiveness of `GMAgent`, demonstrating that our collaborative self-reflection mechanism enables the LLM to focus on crucial information and generate accurate analyses, especially in complex graph structures with rich semantic content.

## A.7 Full Prompt Template

As shown in Table 8 and Table 9, we provide the prompt templates used in repurposing LLMs as graph agents (cf., Section 3.3) and graph-oriented multi-agent collaboration (cf., Section 3.4).

Table 8: Prompt templates used for repurposing LLMs as graph agents on the Arxiv dataset.

| Type | Prompt |
|---|---|
| Task-specific Instruction | **Input:** Given the target PAPER with the effective graph description in the Arxiv dataset, which of the following categories does this PAPER belong to: {*Category List*}. Directly provide the answer to this PAPER's categories.\n\nThe effective graph description of this PAPER is listed as follows: Title: {*Title of PAPER*} Abstract: {*Abstract of PAPER*} Ego Graph Node: {*Ego Graph Node List*} One-hop Neighbor: {*One-hop Neighbor List*} Random Walk: {*Random Walk Paths*}. **Output:** {*Ground-truth Category*}. |
| Querying GPT-4o | I have a question as below: {*Task-specific Instruction Input*} and the answer is {*Task-specific Instruction Output*}. Imagine that you have made the correct choice and proceed with step-by-step reasoning. Your reason needs to incorporate Ego Graph Node, One-hop Neighbor, and Random Walk in the given effective graph description. |
| CoT-based Instruction | **Input:** Given the target PAPER with the effective graph description in the Arxiv dataset, which of the following categories does this PAPER belong to: {*Category List*}. Please think about the categorization in a step-by-step manner and avoid making false associations. Then provide your reason. Using the following format: Answer: {*Answer*}; Reason: {*Reason*}.\n\nThe effective graph description of this PAPER is listed as follows: Title: {*Title of PAPER*} Abstract: {*Abstract of PAPER*} Ego Graph Node: {*Ego Graph Node List*} One-hop Neighbor: {*One-hop Neighbor List*} Random Walk: {*Random Walk Paths*}. **Output:** Answer: {*Ground-truth Category*}; Reason: {*Generated by GPT-4o*}. |

Table 9: Prompt templates used for graph-oriented multi-agent collaboration on the Arxiv dataset.

| Type | Prompt |
|---|---|
| LLM-powered Graph Agent | You are a graph analysis expert, specializing in {*Graph Analysis Perspectives*}. Your task is to analyze the graph based on {*Graph Analysis Perspectives*} within the Arxiv dataset. Consider the following question: {*Task-specific Instruction Input*}. Utilizing your expertise in {*Graph Analysis Perspectives*}, interpret the graph's conditions and emphasize key aspects related to {*Graph Analysis Perspectives*}. Please provide three likely categories as a comma-separated list, arranged from most likely to least likely. For each category, explain your reason. Ensure that your explanation aligns with the answer you provide. Using the following format: Answer: {*Top-3 Likely Category List*} Reason: {*Reason for the Selected Categories*}. |
| Global Summary Agent | Analyze the expert reports related to the target node graph. Your main goal is to evaluate each expert's overall answer based on a holistic confidence score and provide a reason, leading to a cohesive global summary. **1. Confidence Analysis:** Evaluate each expert's entire analysis by assigning a single confidence score on a scale of 1 to 5: 5: Strong confidence; 4: High confidence; 3: Moderate confidence; 2: Low confidence; 1: Poor confidence. For each score, provide a clear and concise justification that reflects the expert's overall depth of knowledge, accuracy, and reliability in their analysis. Ensure that the score represents the expert's report as a whole and not specific sections or categories. **2. Extract Key Insights:** Summarize significant insights relevant to the target node from each expert's report. **3. Global Summary:** Create a unified summary that synthesizes the insights and highlights any critical agreements or controversies. Using the following format: **Confidence Analyses:** One-Hop Neighbors Expert: {*Confidence Score and Reason*} Random Walks Expert: {*Confidence Score and Reason*} Centrality Expert: {*Confidence Score and Reason*} ... **Key Insights:** {*Key Insights*} **Global Summary:** {*Global Summary*}. |
| Collaborative Self-Reflection | Consider the following question: {*Task-specific Instruction Input*}. Based on the question, you have prepared a preliminary analysis: {*Initial Analysis*}. 1. You will receive preliminary analysis from other experts and a synthesized report. Critically review and analyze these insights. 2. If you find aspects of other experts' analysis that are more rational than yours, incorporate these into your analysis for improvement. 3. If you believe your analysis is more scientifically sound compared to others, maintain your stance. Please provide three likely categories as a comma-separated list, arranged from most likely to least likely. For each category, explain your reason. Using the following format: Answer: {*Top-3 Likely Category List*} Reason: {*Reason for the Selected Categories*}. |

