# OpenReview forum: "GMAgent: A Graph-oriented Multi-agent Collaboration Framework for Text-attributed Graph Analysis"
_TMLR — Accepted by TMLR_

### Review · Reviewer_JHwe · 2025-06-08

**Summary Of Contributions:**

1. Formulation of a Graph‐oriented Multi‐Agent Framework. The paper introduces GMAgent, the first multi‐agent collaboration framework that unifies diverse GNN‐based and LLM‐based “graph agents” to jointly analyze Text‐Attributed Graphs (TAGs)  ￼.
2. Effective Model Designs. It presents a suite of novel mechanisms—conflict evaluation among GNN agents, graph‐driven instruction tuning for LLM agents, a role‐play expert recruiting strategy, summary‐report generation, and collaborative self‐reflection—to enable agents to iteratively refine their outputs  ￼.
3. Extensive Experiments. Comprehensive evaluation on five real‐world datasets (node classification and link prediction) demonstrates that GMAgent significantly outperforms state‐of‐the‐art baselines, validating its effectiveness, interpretability, and flexibility

**Audience:**

Yes

**Broader Impact Concerns:**

1. Environmental Footprint: Frequent LLM invocations, especially with GPT‐4o, carry substantial carbon and energy costs, raising sustainability issues.
2. Bias Amplification: Text‐attributed features (e.g., author metadata) may encode societal biases; multi‐agent aggregation could inadvertently reinforce unfair patterns in downstream decisions.
3. Potential Misuse: A powerful TAG‐analysis pipeline could be repurposed for large‐scale social surveillance or automated misinformation generation in networked media.
4. Accessibility Barrier: Dependence on proprietary LLMs limits adoption in resource‐constrained research or industry settings, potentially widening the gap between well‐funded and under‐resourced groups.

**Claims And Evidence:**

Yes

**Requested Changes:**

1. Detailed Efficiency Metrics: Report end‐to‐end wall‐clock runtime and approximate energy consumption for typical datasets, rather than only theoretical complexity.
2. Hyperparameter Ablations: Include experiments varying Tmax, agent counts, and conflict‐threshold parameters to demonstrate robustness.
3. Full Prompt & Tuning Specifications: Provide exact CoT‐based and graph‐description prompts, LoRA ranks, learning rates, and training corpora in an appendix or supplementary material.
4. Larger‐Scale Evaluation: Add at least one large‐scale dataset (e.g., full PubMed citation network) to assess scalability and resource demands.

**Strengths And Weaknesses:**

Strengths
1. Synergistic Integration. By combining GNNs’ global structural learning with LLMs’ local semantic reasoning within a unified multi‐agent loop, the framework leverages complementary strengths in a way not seen in prior work  ￼.
2. Empirical Gains. GMAgent yields an average improvement of 6.67 % on node classification and 3.58 % on link prediction over individual GNN backbones, showing clear quantitative benefits  ￼.
3. Architectural Flexibility. Ablations on varying combinations of GNN agents (e.g., GCN+GAT vs. single TAPE) and LLM agents demonstrate that the framework can be tailored to dataset characteristics without redesigning core components  ￼.

Weaknesses
1. Computational Overhead. Reliance on LLM inference (e.g., GPT‐4o) for conflict scenarios introduces nontrivial latency and energy costs, which may hinder real‐time or large‐scale deployments.
2. Hyperparameter Sensitivity. Key settings—number of self‐reflection iterations (Tmax), number of LLM experts, conflict‐detection thresholds—are presented without sensitivity analysis, leaving questions about stability under different configurations.
3. Reproducibility Gaps. While high‐level descriptions of prompts and LoRA fine‐tuning are provided, exact templates, hyperparameters, and training details are deferred, making precise replication difficult.
4. Scalability Uncertainty. Experiments are limited to mid‐sized academic and citation graphs; the paper does not evaluate performance or resource usage on industrial‐scale TAGs with millions of nodes.

---

> ### Author Response · Authors · 2025-08-12
> **Rebuttal for Reviewer JHwe**
>
> We sincerely thank the reviewer for the constructive comments. Detailed responses are provided below, with revisions marked in BLUE in the revised manuscript.
>
> >**Requested Change 1:** Detailed efficiency metric analysis.
>
> In the revised manuscript, we have (1) verified the impact of different GNN expert combinations and quantified the corresponding proportion of conflict scenarios in **Section 4.3**, highlighting that the majority of instances can be accurately handled by strong GNN experts without invoking expensive LLM-based inferences; and (2) reported the detailed efficiency comparison across different methods in **Section 4.6**, including total runtime and memory usage. Table 1 indicates that GMAgent is a practical and resource-efficient solution for real-time or large-scale deployments.
>
> **Table 1: Efficiency comparison of different methods on Arxiv and Cora.**
> |Dataset|||Arxiv||||Cora||
> |-|-|-|-|-|-|-|-|-|
> |Metric|Micro-F1|Macro-F1|Total Runtime|Memory Usage|Micro-F1|Macro-F1|Total Runtime|Memory Usage|
> |GCN|71.73|51.12|41s|3.1GB|81.87|81.29|2s|0.3GB|
> |GAT|72.24|52.30|3min14s|10.4GB|81.50|81.42|10s|0.5GB|
> |Qwen2-7B|40.25|18.56|334h|15.9GB|58.54|48.35|3h42min|15.6GB|
> |LLaMA3-8B|23.16|11.26|338h|16.7GB|14.76|8.49|3h49min|16.6GB|
> |GMAgent|**78.72**|**59.30**|84h|16.8GB|**85.97**|**85.61**|21min36s|16.7GB|
>
> >**Requested Change 2:** Hyperparameter ablations.
>
> In the revised manuscript, we have included hyperparameter ablation studies on the maximum iteration count $T_{max}$ in **Section 4.5.3**. Table 2 indicates that $T_{max}=2$ as a practical trade-off, balancing predictive accuracy and computational efficiency while ensuring sufficient iterative refinement. Additionally, Section 4.5.1 of the original manuscript already provided experiments varying the number of LLM-based agents. Notably, regarding the conflict‐detection mechanism, our GMAgent does not rely on any fixed threshold parameter. Instead, we adopt an adaptive conflict scenario evaluation strategy based on inconsistent predictions from different GNN experts (cf., Section 3.2.2). This design effectively integrates the global structural modeling capabilities of GNNs, enhancing the robustness of GMAgent.
>
> **Table 2: Influence of varying maximum iteration count.**
> |Dataset|||Arxiv|||Cora|
> |-|-|-|-|-|-|-|
> |$T_{max}$|Micro-F1|Macro-F1|Avg. Inference Time (s)|Micro-F1|Macro-F1|Avg. Inference Time (s)|
> |1|77.01|58.34|3.13|83.21|82.49|1.20|
> |2|**78.72**|**59.30**|6.26|**85.97**|**85.61**|2.39|
> |3|78.17|59.15|9.40|85.12|85.03|3.59|
> |4|77.39|59.03|12.53|84.59|84.71|4.78|
> |5|77.05|58.52|15.67|83.25|84.16|5.98|
>
> >**Requested Change 3:** Full prompt templates and implementation details.
>
> In the revised manuscript, we have (1) provided complete prompt templates in **Appendix A.7**; and (2) included the key fine-tuning specifications in **Section 4.1.4** to ensure precise reproducibility.
>
>
> > **Requested Change 4:** Larger-scale evaluation.
>
> In the revised manuscript, we have added experiments on the large-scale dataset ogbn-products (with 2.45M nodes and 61.86M edges) in **Section 4.2.3** to further verify the scalability and resource demands of our GMAgent. Table 3 demonstrates that GMAgent scales well even on million-node graphs, achieving a practical balance between predictive accuracy and computational efficiency.
>
> **Table 3: Experimental results on ogbn-products.**
> |Dataset|||ogbn-products|||
> |-|-|-|-|-|-|
> |Method|GCN|GAT|Qwen2-7B|LLaMA3-8B|GMAgent|
> |Micro-F1|76.38|73.18|58.02|44.23|**81.65**|
> |Macro-F1|38.07|35.06|38.81|32.57|**81.65**|
> |Total Runtime|4min49s|23min25s|340h|367h|81h|
> |Memory Usage|15.3GB|23.1GB|15.8GB|16.6GB|16.7GB|

---

> > ### Author Response · Authors · 2025-08-13
> > **Correction to Table 3**
> >
> > We corrected the results in Table 3 due to a typographical error in the previous version.
> >
> > **Table 3: Experimental results on ogbn-products.**
> > |Dataset|||ogbn-products|||
> > |-|-|-|-|-|-|
> > |Method|GCN|GAT|Qwen2-7B|LLaMA3-8B|GMAgent|
> > |Micro-F1|76.38|73.18|58.02|44.23|**81.65**|
> > |Macro-F1|38.07|35.06|38.81|32.57|**44.27**|
> > |Total Runtime|4min49s|23min25s|340h|367h|81h|
> > |Memory Usage|15.3GB|23.1GB|15.8GB|16.6GB|16.7GB|

---

### Review · Reviewer_Mq2L · 2025-07-06

**Summary Of Contributions:**

This paper introduces GMAgent, a framework that integrates GNNs and LLMs into a multi-agent collaboration system for analyzing text-attributed graphs (TAGs). It leverages GNNs for structural understanding and LLMs for semantic reasoning, selectively applying LLM-based collaboration in conflict cases. Extensive experiments across five datasets demonstrate significant performance gains over state-of-the-art methods.

**Audience:**

Yes

**Broader Impact Concerns:**

There is no concerns on ethical implication in this work.

**Claims And Evidence:**

Yes

**Requested Changes:**

1. Clarify and evaluate the graph textualization mechanism. Provide ablation studies or user studies that compare different graph-to-text strategies (e.g., with or without random walks or structural context) to demonstrate how effectively LLMs can comprehend graph information through text alone.
2. Detail the conflict scenario selection policy. Elaborate on how the selection of conflict scenarios is performed and provide a quantitative analysis of its impact (e.g., how often conflicts are detected, how conflict filtering affects downstream performance). Consider alternative strategies such as learned selectors or threshold-based confidence aggregation.

**Strengths And Weaknesses:**

Strengths:
1. The paper is well-organized and presents a clear and intuitive framework that is easy to follow.
2. The integration of GNNs and LLMs in a role-based, multi-agent collaboration framework is novel and addresses key limitations in previous methods.
3. The framework is validated through experiments, showing substantial improvements in both node classification and link prediction tasks across diverse datasets.

Weaknesses:
1. The paper assumes LLMs can understand graph structures through textual descriptions (e.g., one-hop neighbors, and random walks), but does not evaluate or justify the effectiveness of this graph-to-text transformation. This modality shift may introduce ambiguity or information loss, and warrants further investigation. It is unclear how LLM can understand the graph description based on text. The mechanism of how much LLM can effectively comprehend the give graph information requires separate studies.
2. The selection mechanism for triggering LLM-based collaboration in conflict scenarios is only briefly described. It remains unclear how this policy would generalize to more complex or ambiguous cases in real-world applications, and whether it introduces bias or inefficiency.

---

> ### Author Response · Authors · 2025-08-12
> **Rebuttal for Reviewer Mq2L**
>
> We sincerely thank the reviewer for the insightful comments. Below, we provide detailed responses. Corresponding revisions have been marked in BLUE in the revised manuscript.
>
> > **Weakness 1:** Clarify the graph textualization mechanism.
>
> Our graph textualization mechanism explicitly integrates one-hop neighbors and random walks to provide comprehensive descriptions of graph structures for LLM input. Specifically, one-hop neighbors facilitate the understanding of local connectivity and feature distributions [1-3], offering fine-grained insights into node attributes. In contrast, random walks serve as a dynamic strategy to capture global structural patterns and high-order relationships [4-6], highlighting the diversity of connectivity and paths.
>
> **Reference**
> [1] Inductive Representation Learning on Large Graphs, NeurIPS, 2017.
> [2] Semi-supervised Classification with Graph Convolutional Network, ICLR, 2017.
> [3] Relation Learning on Social Networks with Multi-modal Graph Edge Variational Autoencoders, WSDM, 2020.
> [4] Anonymous Walk Embeddings, ICML, 2018.
> [5] Higher-order Attribute-enhancing Heterogeneous Graph Neural Networks, TKDE, 2021.
> [6] WalkLM: A Uniform Language Model Fine-tuning Framework for Attributed Graph Embedding, NeurIPS, 2023.
>
>
> >**Weakness 2:** Clarify the conflict scenario selection policy.
>
> Our conflict scenario selection policy effectively identifies instances where different GNN experts yield inconsistent predictions, while LLM-based collaboration can be beneficial. Specifically, for each input instance (e.g., a node in classification), if all GNN experts produce the same inference result, the result is directly accepted. Otherwise, the instance is treated as a conflict scenario and passed to LLMs for collaborative reasoning with the corresponding GNN outputs. This output-level detection is model-agnostic, enabling straightforward extension to settings with more experts, heterogeneous architectures, or different domains, while avoiding reliance on any single model's bias.
>
>
> >**Requested Change 1:** Evaluate the graph textualization mechanism.
>
> In the revised manuscript, we have added ablations on different graph description variants in **Appendix A.5**. Table 1 validates the effectiveness of our textualization mechanism in enabling LLMs to comprehend and analyze graphs. Additionally, Section 4.4 of the original manuscript demonstrates that fine-tuning LLM-based agents via these descriptions significantly enhances their performance in graph analytical tasks.
>
> **Table 1: Influence of different graph description strategies.**
> |Dataset|Arxiv||Cora||
> |-|-|-|-|-|
> |Metric|Micro-F1|Macro-F1|Micro-F1|Macro-F1|
> |Base|75.53|55.09|81.29|80.92|
> |+ Ego Graph Node|76.21|56.85|82.34|82.01|
> |+ One-hop Neighbour|77.46|58.02|84.15|83.74|
> |GMAgent|**78.72**|**59.30**|**85.97**|**85.61**|
>
> >**Requested Change 2:** Quantitative analysis of the conflict scenario selection policy.
>
> In the revised manuscript, we have (1) provided a clearer description of the selection policy in **Section 3.2.2**; (2) included detailed conflict ratio and performance analysis in **Section 4.3**; and (3) added an efficiency comparison in **Section 4.6**. Tables 2-3 highlight that integrating any GNN agent combinations leads to a certain proportion of conflict scenarios, with an average conflict ratio of 13.29% across all combinations and a range of 11.61%-15.96%. This indicates that the conflict evaluation mechanism filters most consistent predictions as final outputs, alleviating the frequent call to costly LLMs. Resolving the remaining conflicts via LLM collaboration yields potential improvements in predictive accuracy, with an average Micro-F1 improvement of 6.70% over the corresponding single-GNN baselines on Arxiv. For future work, we will explore learned selectors and threshold-based confidence aggregation to further adapt the policy to high-noise or low-confidence scenarios.
>
> **Table 2: Performance (%) comparison of different methods on Arxiv.**
> |Micro-F1|GCN|GAT|GraphSAGE|TAPE (GCN)|
> |-|-|-|-|-|
> |GCN|71.73|**78.72**|77.24|76.89|
> |GAT|**78.72**|72.24|77.01|77.45|
> |GraphSAGE|77.24|77.01|71.45|76.36|
> |TAPE (GCN)|76.89|77.45|76.36|74.36|
>
>
> **Table 3: Conflict ratio (%) comparison of different methods on Arxiv.**
> |Conflict Ratio|GCN|GAT|GraphSAGE|TAPE (GCN)|
> |-|-|-|-|-|
> |GCN|-|11.61|11.71|14.41|
> |GAT|11.61|-|11.71|14.34|
> |GraphSAGE|11.71|11.71|-|15.96|
> |TAPE (GCN)|14.41|14.34|15.96|-|

---

### Review · Reviewer_cjkt · 2025-07-29

**Summary Of Contributions:**

The paper proposes a multiagent system solution for tasks related to text-attributed graphs. The authors therefore propose to combine different agents, either being a graph neural network (GNN) or an LLM. Individual prompts wrap the predictions of used GNNs and LLMs, and an agent collaboration mechanism based on so-called collaborative self-reflection and a performance report is proposed.

The authors evaluate their method on node classification and link prediction tasks for datasets arxiv, cora, imdb, pubmed, dblp. Various baselines from competing GNN and LLM approaches are chosen.

The results show that the proposed approach surpasses all baselines for both tasks. Ablations also show that some agent combinations work better than other for the proposed system.

**Audience:**

Yes

**Broader Impact Concerns:**

-

**Claims And Evidence:**

No

**Requested Changes:**

- Related work on Multiagent systems: Recent work [1] should be mentioned. In addition, where do you situate your multiagent collaboration approach in? Is it completely new in the multiagent / expert advice literature? If so, why can't we reuse other combination strategies? Quite some are cited, but only a general statement is given.
- Multiagent baselines: Please add recent multiagent baselines for text-attributed graph tasks. This could be either only based on LLMs [1] or mentioned close competitors [2] or something similar but representative. Otherwise a strong argumentation is necessary why this is not needed.
- Discussion of ablation results: Please add deeper discussion on the agent ablation in 4.3 wrt if your system is able to best combine the available agents. Is it that the systems makes more mistakes when combined some agents, which could hypothetically achieve better results?

[1] MARK: Multi-agent Collaboration with Ranking Guidance for Text-attributed Graph Clustering, https://aclanthology.org/2025.findings-acl.314.pdf
[2] Can Graph Learning Improve Planning in LLM-based Agents?, https://arxiv.org/pdf/2405.19119

**Strengths And Weaknesses:**

# Strengths
- The approach is understandable and sensible. The methods for wrapping of individual GNN or LLM approaches and their combination become clear from the manuscript.
- The evaluation uses a diverse set of individual GNN and LLM approaches

# Weaknesses
- The evaluation lacks a multiagent baseline of any sort. There is relevant related work which could be implemented at this point (or simple combination strategies).
- Presentation could/should be improved: The introduction would benefit from giving explicit examples of relevant tasks, not only mentioning text-attributed graphs in general.

---

> ### Author Response · Authors · 2025-08-12
> **Rebuttal for Reviewer cjkt**
>
> We sincerely appreciate the reviewer's valuable comments. Please find our point-by-point responses below. The corresponding revisions are highlighted in BLUE in the revised manuscript.
> > **Weakness 1:** Add related work and baselines on multi-agent systems.
>
> In the revised manuscript, we have (1) incorporated the recent work MARK [1] into Section 2.2; (2) added MARK [1] into our experimental comparison in Section 4.2. Particularly, since MARK was officially published at ACL in June 2025 after our submission deadline, it was not included initially. Furthermore, although [2] explores the integration of GNNs and LLMs within a multi-agent framework, it focuses on task planning, rather than text-attributed graph tasks. Therefore, we discuss its insights in our related work section but do not include it as a direct baseline. Table 1 shows that our GMAgent offers an effective and flexible multi-agent collaboration framework that integrates the global structural learning abilities of GNNs and the local semantic richness of LLMs, enabling broad adaptability across diverse graph analytical tasks.
>
> Additionally, our proposed GMAgent belongs to the category of collaborative multi-agent systems, where agents share a common graph analysis objective and collaborate through iterative self-reflection. Notably, GMAgent is the first multi-agent framework that integrates the global structural learning capabilities of GNNs with the local semantic richness of LLMs, enabling flexible collaboration across various graph analytical tasks. In contrast, MARK identifies uncertain nodes through graph clustering and employs LLM-based multi-agent collaboration to generate guidance, which inherently constrains its generalization to broader graph tasks like multi-label node classification (e.g., IMDB) or link prediction (e.g., PubMed). While our current strategy adopts collaborative self-reflection, GMAgent can seamlessly support other collaboration strategies, such as real-time role adaptation or debate-based coordination. We consider this a promising direction for future work to further enhance efficiency and adaptability.
>
> **Table 1: Experimental results (%) on three datasets for node classification.**
> |Dataset|Arxiv||Cora||IMDB||
> |-|-|-|-|-|-|-|
> |Metric|Micro-F1|Macro-F1|Micro-F1|Macro-F1|Micro-F1|Macro-F1|
> |MARK|55.42|44.70|72.35|69.32|-|-|
> |GMAgent|**78.72**|**59.30**|**85.97**|**85.61**|**74.36**|**66.82**|
>
> > **Weakness 2:** Add task-specific examples in the introduction.
>
> In the revised manuscript, we have added explicit examples of relevant tasks (e.g., node classification and link prediction).
>
> > **Requested Changes 1-2:** See the above **Weakness 1** and **Weakness 2**.
>
> > **Requested Change 3:** Deeper discussion on the GNN-based graph agent ablation.
>
> In the revised manuscript, we have included the detailed conflict ratio and performance analysis in Section 4.3. Tables 2-3 highlight that integrating any GNN agent combinations leads to a certain proportion of conflict scenarios, with an average conflict ratio of 13.29% across all combinations and a range of 11.61%-15.96%. This indicates that the conflict evaluation mechanism filters most consistent predictions as final outputs, alleviating the frequent call to costly LLMs. Resolving the remaining conflicts via LLM collaboration yields potential improvements in predictive accuracy, with an average Micro-F1 improvement of 6.70% over the corresponding single-GNN baselines on Arxiv. Furthermore, we observe that performance varies across different combinations of GNN agents. For example, GCN+GAT outperforms GCN+TAPE (GCN) by 2.38% in Micro-F1 on Arxiv.
>
>
> **Table 2: Performance (%) comparison of different methods on Arxiv.**
> |Micro-F1|GCN|GAT|GraphSAGE|TAPE (GCN)|
> |-|-|-|-|-|
> |GCN|71.73|**78.72**|77.24|76.89|
> |GAT|**78.72**|72.24|77.01|77.45|
> |GraphSAGE|77.24|77.01|71.45|76.36|
> |TAPE (GCN)|76.89|77.45|76.36|74.36|
>
>
> **Table 3: Conflict ratio (%) comparison of different methods on Arxiv.**
> |Conflict Ratio|GCN|GAT|GraphSAGE|TAPE (GCN)|
> |-|-|-|-|-|
> |GCN|-|11.61|11.71|14.41|
> |GAT|11.61|-|11.71|14.34|
> |GraphSAGE|11.71|11.71|-|15.96|
> |TAPE (GCN)|14.41|14.34|15.96|-|

---

### Decision · Action_Editor_51CW · 2025-09-30

**Recommendation:** Accept with minor revision

**Audience:**

Yes

**Audience Explanation:**

This work will attract TMLR's audience of graph neural networks, LLMs, and multi-agent systems.

**Claims And Evidence:**

Yes

**Claims Explanation:**

The paper introduces GMAgent, a multi-agent framework that integrates GNNs and LLMs for analyzing text-attributed graphs. Its design - featuring role-based collaboration, self-reflection, and report generation - is well-motivated and effectively executed, with strong performance across multiple datasets and tasks. The combination of GNNs’ structural reasoning with LLMs’ semantic capabilities is impactful. Remaining concerns focus on the justification and validation of graph-to-text transformation for LLMs, which may introduce redundancy and information loss, as well as the limited discussion of the selection mechanism for LLM collaboration. After a careful reading, I find the work interesting and solid, and believe these issues can be addressed after revision. I therefore recommend acceptance with minor revision.